# Uprooting defects to enable high-performance III–V optoelectronic devices on silicon

Youcef A. Bioud[1]*, Abderraouf Boucherif[1]*, Maksym Myronov[2], Ali Soltani[1], Gilles Patriarche[3], Nadi Braidy[1,4], Mourad Jellite[1], Dominique Drouin[1] & Richard Arès[1]*

The monolithic integration of III-V compound semiconductor devices with silicon presents physical and technological challenges, linked to the creation of defects during the deposition process. Herein, a new defect elimination strategy in highly mismatched heteroepitaxy is demonstrated to achieve a ultra-low dislocation density, epi-ready Ge/Si virtual substrate on a wafer scale, using a highly scalable process. Dislocations are eliminated from the epilayer through dislocation-selective electrochemical deep etching followed by thermal annealing, which creates nanovoids that attract dislocations, facilitating their subsequent annihilation. The averaged dislocation density is reduced by over three orders of magnitude, from $\sim 10^8$ cm$^{-2}$ to a lower-limit of $\sim 10^4$ cm$^{-2}$ for 1.5 μm thick Ge layer. The optical properties indicate a strong enhancement of luminescence efficiency in GaAs grown on this virtual substrate. Collectively, this work demonstrates the promise for transfer of this technology to industrial-scale production of integrated photonic and optoelectronic devices on Si platforms in a cost-effective way.

[1] Laboratoire Nanotechnologies Nanosystèmes (LN2)—CNRS UMI-3463, Institut Interdisciplinaire d'Innovation Technologique (3IT), Université de Sherbrooke, 3000 Boulevard Université, Sherbrooke J1K OA5 QC, Canada. [2] Department of Physics, University of Warwick, Coventry CV4 7AL, UK. [3] Centre for Nanoscience and Nanotechnology, CNRS, Université Paris-Sud, Université Paris-Saclay, Route de Nozay, 91460 Marcoussis, France. [4] Department of Chemical and Biotechnological Engineering, Université de Sherbrooke, 2500 Boul. de l'Université, Sherbrooke, QC J1K 2R1, Canada. *email: Y.Bioud@USherbrooke.ca or A.Boucherif@USherbrooke.ca or Richard.Ares@USherbrooke.ca

Developing inexpensive, high-performance epitaxial devices with the hope of achieving widespread market adoption, depends critically on production costs, and ultimate material quality. Any process that meets these challenges successfully provides an attractive solution for a wide range of leading-edge applications, in different areas, including energy, photonics, electronics, communications, and health care. The last decade has witnessed considerable progress in many advanced technologies based on mismatched hetero-epitaxial semi-conductors, such as solar cells, LED, and laser sources[1–6]. As an example, a GaInP/GaAs//GaInAsP/GaInAs four-junction solar cell achieves a record efficiency of 46% exceeding those of other technologies[7]. This structure must however, combine two tandem cells grown on different lattice constant substrates that are subsequently bonded together. This application demonstrates how powerful heterogeneous material integration can be. However, the monolithic integration of high-quality mismatched semi-conductors remains difficult due to the large difference in the lattice constant and thermal expansion coefficients between the epitaxial materials and hetero-substrates[8]. During growth, the strain, which is induced in the epitaxial layer will be either stored as strain energy in the film or accommodated by a network of misfit dislocations at the interface above a certain critical thickness, depending on the lattice mismatch[9,10]. Misfit dislocation segments are always accompanied by a high density of threading dislocations (TD) extending to the surface[11]. TD are the most undesirable, because they commonly penetrate active device regions and generate a variety of detrimental effects, such as non-radiative recombination centers for carriers, reducing their mobility and lifetime[12–14], optical birefringence[15], or parasitic current leakage paths[16]. Devices like photodetectors[17,18], multi-junction solar cells (MJSC)[19], microprocessors[20], modulators[21], based on Ge-on-Si (001) technology must tackle these issues in order to achieve industry standards. To this end, several approaches have been proposed for reducing the TD density, such as compositional grading[22], cyclic annealing[23,24], epitaxial lateral overgrowth[25], selective area depositions[26], 3D heteroepitaxy[27], virtual graded layers[28], mesa structuring[29], compliant sub-strates[30], and ion-implanted substrates[31]. Despite their feasibility and originality, many of these techniques are limited to small-scale processes and require the use of expensive and complex processing technologies[32]. In addition, very thick graded buffers are causing substantial wafer bow, which makes the epi wafers unsuitable for further wafer scale devices fabrication using lithography and other processes. Despite the implementation of mitigation measures, in a number of these devices, the TD density remains high (~$10^6$ cm$^{-2}$)[33]. Alternatively, a new approach called aspect ratio trapping (ART) has been proposed as one of the strategies that has the potential to completely annihilate dislocations[17,34]. Through the use of substrate patterning, the ART method favors the termination of TD at free surfaces on the side facets of the patterns, leading to a confinement of dislocations in lateral sectors or their termination[35]. Although the ART method shows good potential for reducing the TD density in active device regions, nanoscale-patterning lithography, and the discontinuous characteristics of the resulting Ge/Si films, can limit their application.

This work aims at providing a reliable approach for achieving very low dislocation density, by tuning the motion of dislocations in relaxed Ge epilayers grown on Si substrates through self-assembling nanovoids. The approach proposed herein involves only a basic, industry standard two-step process (electrochemical etching and thermal annealing); which we expect will likely have a small impact on the overall processing costs of a device. This process enables two things; the movement and rearrangement of dislocations through the porosified material and the re-solidification of the Ge epilayer. Advantages of this technique include low cost, large surface area, and compatibility with microfabrication facilities and other processing steps. The key cost-reducing aspects of this process flow are the intrinsic com-patibility of each process step with large area semiconductor mass production. Both electrochemical porosification and thermal annealing are standard, scalable techniques that are already well integrated within an industrial fully automatic environment. Our process requires no rare or special material or treatment, nor does it take long processing times to perform. Current production tools for this process are also readily available and well established.

One area in which this approach can provide a significant opportunity is the fabrication of virtual substrates for III–V MJSC on Si[36]. As a result, significant cost savings would be made possible if the bulk Ge substrates could be replaced with our virtual analogs consisting of micron-thick, or even less, Ge buffers grown on Si wafers[37]. The cost reduction per solar cell can be as high as 75% when the much lower prices and larger areas of Si wafers are considered[38]. Additionally, this approach would offer a viable path towards the monolithic integration of mismatched Ge/Si in CMOS circuits and overcome their imminent perfor-mance limitation[39,40].

This work shows that dislocation engineering using nanovoids in the heteroepitaxy of Ge/Si can be a plausible path for mono-lithic integration of high-quality GaAs on Si platform. The innovation here is to use this strategy in a mismatched Ge/Si structure to reveal new beneficial phenomena for reducing the TD density. Possible mechanisms responsible for TD density reduc-tion that lead to either annihilation or fusion through nanovoids are discussed. The results show clear evidence that introducing nanovoids increases recombination probabilities of TD, drasti-cally reducing their presence within the Ge layer. As a result, cathodoluminescence (CL) measurements shows an increased CL signal from the GaAs layer on the engineered Ge/Si substrate, in stark contrast to the CL signal from the GaAs layer on a con-ventional Ge/Si substrate. Such an enhancement is attributed to the role of nanovoids in reducing the TD density.

## Results

**Ge/Si virtual substrate: design and fabrication.** In this work, a Ge/Si substrate is treated with a post epitaxial process to create a region with a high density of nanovoids within the Ge/Si struc-ture, which acts as a free surface that attracts dislocations, facil-itating the subsequent annihilation of their threading arms. The process is illustrated schematically in Fig. 1. Nanovoids are formed in the Ge layer as well as in the Si substrate by electro-chemical porosification, followed by thermal annealing, creating a new configuration referred to as nanovoid-based Ge/Si virtual substrate (NVS).

As a first step, Ge/Si (001) samples with high TD density are anodically porosified by the bipolar electrochemical etching (BEE) technique previously that was described in refs. [41–43]. Contrary to bulk Ge porosification, in which, the layer structure exhibits uniformly distributed mesopores with controllable nanostructure and size[44–46], dislocated Ge/Si substrates show selectively distributed mesopores. Typical SEM views at low and high magnifications of the mesoporous Ge/Si layer prepared under 1.5 mA/cm$^2$ are presented in Fig. 2a, b, respectively. Large pores in the range of 50 nm are formed near the dislocation core sites, which essentially reveal the emergent point of the dislocations at the surface[47]. This is reflected in the cross-sectional view of a thick porous Ge (PGe) layer with weakly branched pores that tend to follow threading dislocation cores, as shown in Fig. 2c. Calibration of porous layer parameters, such as

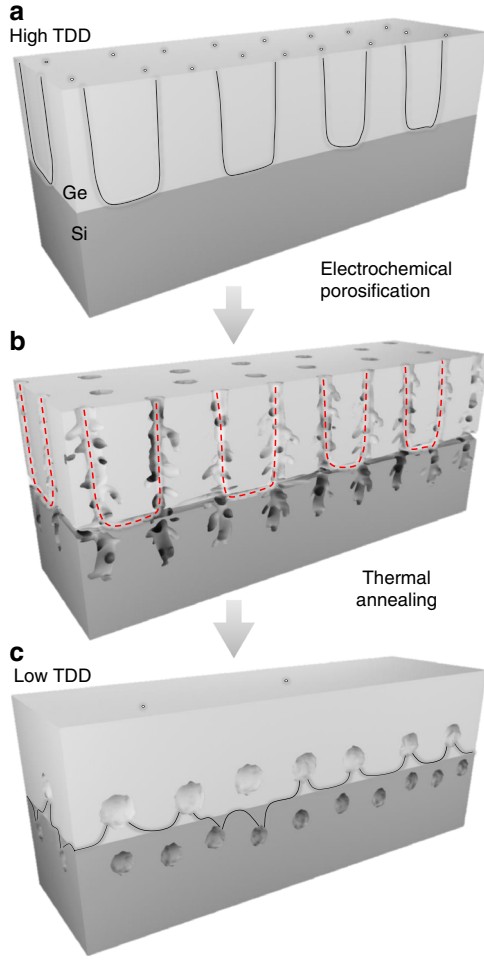

**Fig. 1** Schematic illustration of the proposed approach. **a** Bulk Ge film growth on (001) Si substrate. **b** Dislocation-selective electrochemical deep etching to form porous nanostructures from the bulk-grown Ge film and the Ge/Si interface. The dotted line shows weakly branched pores that follow threading dislocation cores. **c** Thermal annealing of the nanoporous structure to transform it into nanovoids that attract dislocations, facilitating their subsequent annihilation. The resulting configuration is referred to as nanovoid-based Ge/Si virtual substrate (NVS)

the etching rate, has been done by varying the current density in the electrochemical process of bulk and dislocated Ge in Fig. 2g. The etch rate ratio between dislocated and bulk regions shows very high values up to 6. Beyond 3.7 mA/cm², porous Si layer was obtained by hydrofluoric acid (HF) crossing dislocations. Figure 2d shows a dark-field transmission electron microscopy (TEM) image of PGe/Si formed using 4 mA/cm² for the electrochemical etching for a period of 30 min. The porosification front has reached the Ge/Si heterointerface, by following primarily threading segments. The high density of misfit dislocations within the interface has also favored a significant etching effect in that area. A porous Si layer of 200 nm is formed after 35 min, even before the complete porosification of the Ge layer (see Fig. 2e, f).

Figure 2h presents schematically the energy band diagrams of p-doped Ge with the HF electrolyte, in the case of bulk and dislocated substrates, showing the surface energy band bending in each case. At thermodynamic equilibrium, the Fermi energy ($E_f$) in Ge is aligned with the equilibrium energy ($\mu$) of the electrolyte, causing the formation of an electrical double layer from the electrolyte side and the surface charge region (SCR) within the

electrode. As a result, there is bending of the energy bands close to the Ge surface and a Schottky barrier ($e\Delta\Phi_{SCL}$) that inhibits hole injection across the interface is created. When a positive bias is applied to the Ge electrode, the SCR is reduced and valence band holes begin to accumulate at the interface. The holes participate in the electrochemical etching reactions according to:

$$Ge + 6HF + 4h^+ \rightarrow H_2GeF_6 + 4H^+ \qquad (1)$$

The preferential anodic etching through dislocations could manifest itself as a local effect. In fact, the impurity gettering by the dislocation core increases doping concentration locally, which is reflected by the displacement of the Fermi level[48]. As a result, a lower energy barrier $e\Delta\Phi_{SCL}$ increases Ge dissolution during anodization, thus accelerating pore formation around dislocations, which act as selective nucleation sites. Another effect, which could stimulate this localized dissolution, can be seen from a structural viewpoint, in which the lattice is locally distorted for a distance of a few atoms around the dislocations cores. As a result of the stress field generated by the deformation, the lattice elements dissolve more easily near the dislocation cores than in stress-free, undeformed areas[49]. Such preferential porosification significantly enhances the atomic mobility allowing preferential diffusion paths for voids and dislocations during annealing.

The next step consists of thermally induced structural reorganization of the porous nanostructure. The morphological change of PGe/Si during annealing is based on thermally activated diffusion of surface atoms to stable positions following a surface diffusion mechanism, for which atoms seek to complete a maximum of covalent bonds until their outer energy level is full[50–53]. The driving force for the spherodization of pores results from the system's tendency to minimize its total surface energy. Such a change has been previously observed following Rayleigh instabilities[54,55]. Depending on the annealing temperature, different void shapes in the Ge layer are obtained (Supplementary Fig. 1), which are in a good agreement with the theory describing the Oswald ripening phenomenon[56,57], in which smaller voids tends to merge together in order to create larger voids in a process that is driven by reducing the total surface energy. The restoration of the bulk order in the annealed PGe matrix is confirmed by $\mu$-Raman spectroscopy (Supplementary Fig. 1). The void formation in the Ge layer occurs in an isotropic manner, thanks to the dislocation-selective electrochemical etching, and in an anisotropic manner in the Si substrate (Fig. 2a). Our observations show that this architecture is effective in reducing the TD density.

**Dislocation annihilation mechanisms and crystal quality.** The dislocation annihilation mechanism is a two-step process in which multiple dislocations are attracted to the same nanovoid, which then leads to enhanced probability of annihilation of their threading arms. In order to investigate the presence of misfit and TD (MDs and TDs) throughout the NVS and study their interactions with nanovoids, detailed structural characterization was performed by scanning transmission electron microscopy (STEM) and TEM. High magnification STEM view in dark field (DF) and bright field (BF) shows clearly the overlap dislocation-voids as can be seen in Fig. 3b, c. Different processes may come into play in the dislocation pinning mechanism including; gliding[58] and climbing processes[59,60], which depend on temperature, strain, obstacle specifications (type, diameter, and spacing)[61–63] and other critical parameters[64]. For Ge/Si (001), dislocation arms are located on the (111) and the (11−1) slip systems. An initially perfect 60° edge dislocation dissociates into two separated Shockley partials (e.g., 60° dislocations dissociate into 30° and 90° partial dislocations), in which, the energy state of the sum of the partials is less than the energy state of the original

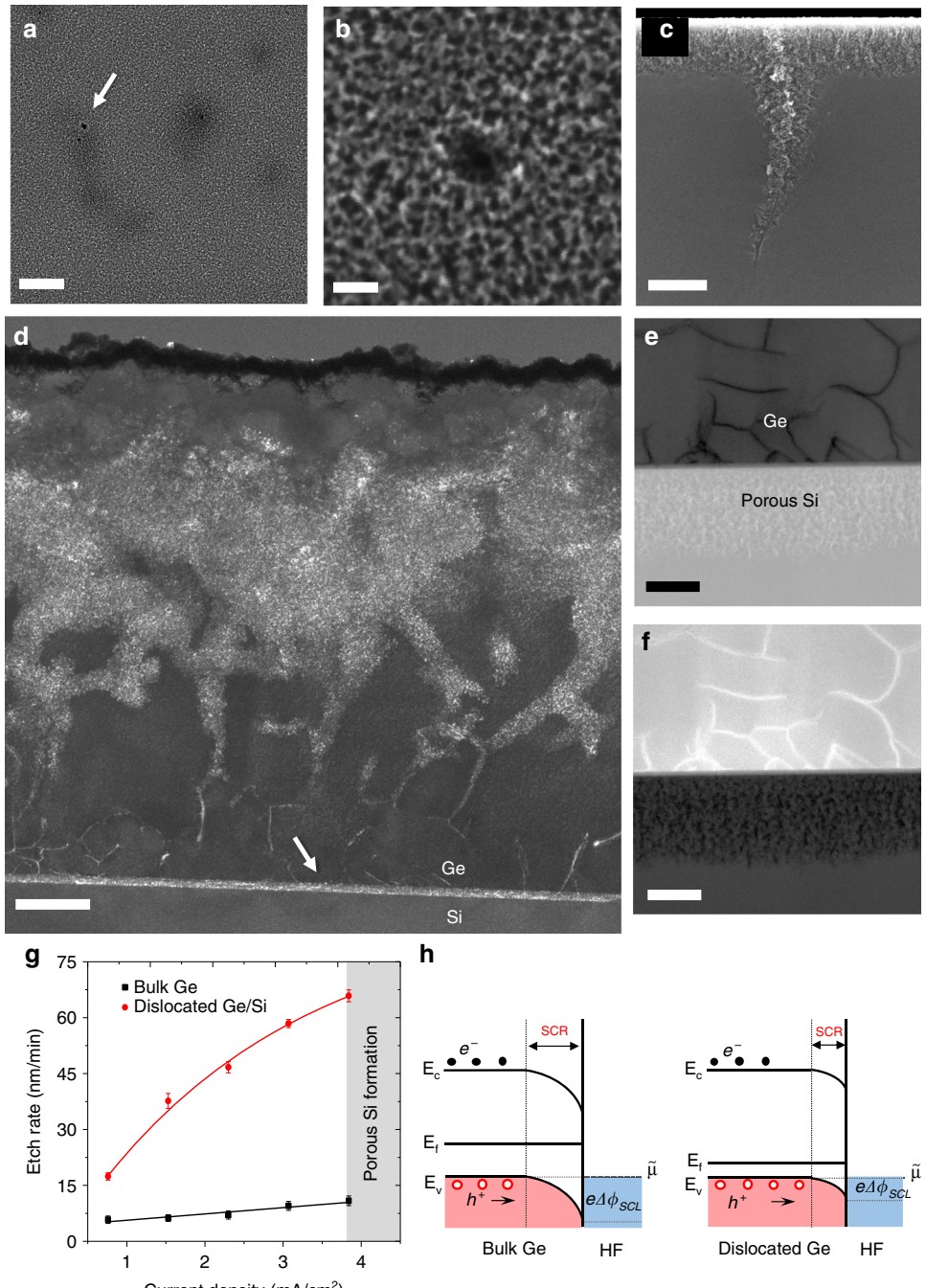

**Fig. 2** Dislocation-selective electrochemical deep etching. **a** Planar-view scanning electron microscopy (SEM) image of porous Ge/Si layer formed at 1.5 mA/cm². The arrow shows large pores formed near the dislocation core sites, which essentially reveal the emergent point of the dislocations at the surface as seen in the high-magnification SEM image (**b**). Scale bars 300 nm (**a**), 30 nm (**b**). **c** Cross-section SEM view showing preferential etching through dislocations. Scale bar 200 nm. **d** Dark-field transmission electron microscopy (DFTEM) image of porous Ge/Si formed using a current density of 4 mA/cm² for 30 min shows a thick porous Ge layer with weakly branched pores crossing threading dislocation cores up to full uprooting of misfit dislocations, as indicated by the arrow. Scale bar 200 nm. Bright field BF-scanning transmission electron microscopy (**e**) and DF-scanning transmission electron microscopy (STEM) (**f**) micrographs showing the formation of porous Si obtained by HF crossing dislocations at 4 mA/cm² during 35 min. Scale bars: 100 nm. **g** Etching rate evolution versus the current density of bulk Ge and dislocated Ge/Si layer. Error bars represent standard deviation of the average etching rate. The energy band diagrams of p-doped Ge with HF electrolyte under a positive bias, in the case of bulk and dislocated substrates, showing the surface energy band bending in each case. **h** The displacement of the Fermi level in the dislocated Ge is due to a rise of the doping concentration locally around dislocations cores. The illustration is not based on any calculations. Source data are provided in the Source Data file

dislocation[65,66]. The attraction and detachment process between dislocations and voids depends on the void parameters and the distance between the partials[67]. For the smaller voids, the partials strength dominates the obstacle strength. Consequently, the dislocations cut through the small voids without being trapped. For the larger voids, the obstacle strength dominates, causing the dislocation to be pinned by the voids and to bow under the internal shear stress. In addition, the void induces a stress field in its surroundings, which strongly influences the dislocation passage depending on the geometry of the interaction[68–70]. A short

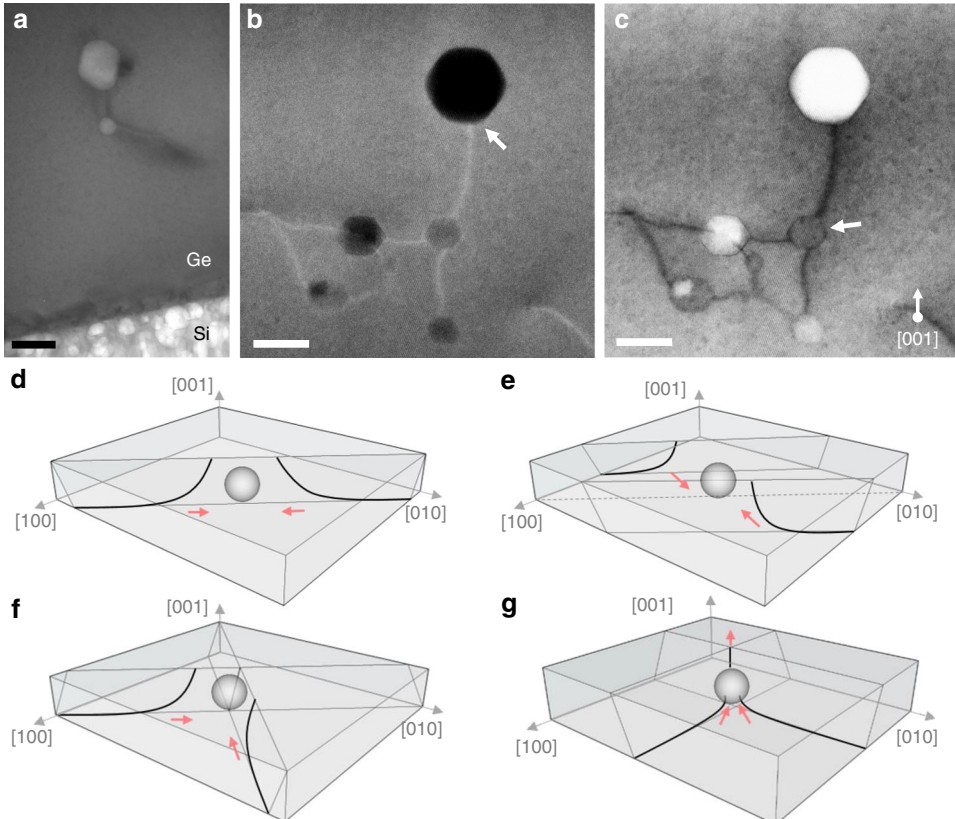

**Fig. 3** Possible mechanisms responsible for dislocation annihilation. **a** Low magnification transmission electron microscopy (TEM) image close to the Ge/Si interface showing the presence of the voids in the Ge layer as well as in the Si substrate. Scale bar 100 nm. Scanning transmission electron microscopy (STEM) images present different interactions between dislocations and voids in dark field (**b**) and bright field (**c**). Scale bars 50 nm. The gliding dislocations in the Ge layer shows slight curvature to join the void area until its complete annihilation. Possible processes for annihilation of threading dislocations at the void surface by interaction of several threading segments from: the same slip system (**d**), the parallel slip system (**e**), by a combination of glide and slip motion (**f**), and by fusion (**g**). Source data are provided in the Source Data file

void spacing induces—high-stress concentration and the resolved shear stress and consequently the obstacle strength[71]. Therefore, for large voids that are widely spaced out, dislocation can bypass the obstacle, as the stress concentration formed around the voids is low. Indeed, voids with adequate dimensions and spacing create a stable, energetically favorable configuration for the dislocations, effectively pinning them.

For TDs annihilation, the arrow in Fig. 3b indicates the location where a dislocation segment terminates on the nanovoid with no propagation beyond. A reasonable explanation for the TD termination is that two dislocations or more with opposite Burgers vectors having the same magnitude have reached the void. From Frank's rule of conservation, the net Burgers vector resulting from this overlap must be zero[72]. Possible reactions between TD in heteroepitaxial cubic semiconductors have been presented including fusions, complete, and half-annihilation reactions. As demonstrated in Fig. 3d, e, threading segments from either different dislocation sources on the same slip plane (111) (e.g., $a/2[1{-}10]$ and $a/2[{-}110]$), or on the parallel slip system (e.g., $a/2[01{-}1]$ and $a/2[0{-}11]$) can annihilate. In addition, by thermally activating both glide and climb of the threading segments, annihilation may be achieved from different dislocation sources on intersecting slip systems as shown in Fig. 3f (e.g., $a/2[101]$ and $a/2[{-}10{-}1]$)[71]. The fusion of threading segments is also an energetically favorable reaction, in which two dislocations merge to form a single one generating a Y-shaped defect as highlighted by the arrow in Fig. 3c and schematized in

Fig. 3g. For example, fusion is favorable between two TDs with Burgers vectors (e.g., $a/2[10{-}1]$ and $a/2[011]$) leading to a single TD with Burgers vectors $a/2[110]$[73].

These different interactions between neighboring dislocations are characterized by the physically prescribed characteristic interaction distance, which corresponds to the distance at which the interaction force between dislocations becomes larger than the friction resistance of the lattice[71]. Due to the attractive force created by the population of smaller voids, the approach distance between two TDs with different Burgers vectors becomes low enough, so that the TDs begin to glide together and annihilate or fuse. Figure 4 proposes a descriptive sketch to explain the phenomena of the annihilation of dislocations in Ge epilayer grown on Si substrate by introducing nanovoids. For the sake of clarity, only one {111} glide plane is drawn. The heat treatment stimulates the propagation of dislocation loops along glide planes (Fig. 4b). The dislocations with opposite Burger vectors move easily and react with each other (Fig. 4c), since it can occur in the same glide plane {111}, without the need of any extra point defect supersaturation for climbing. Thus, the threading components disappear (Fig. 4d). These mechanisms give a reasonable explanation for the dislocations reduction by introducing nanovoids in heteroepitaxial diamond films. The characteristic interaction distance over which two dislocations interact becomes higher than that in the conventional structure without voids, which in turn increases their recombination probability. However, the full process could be

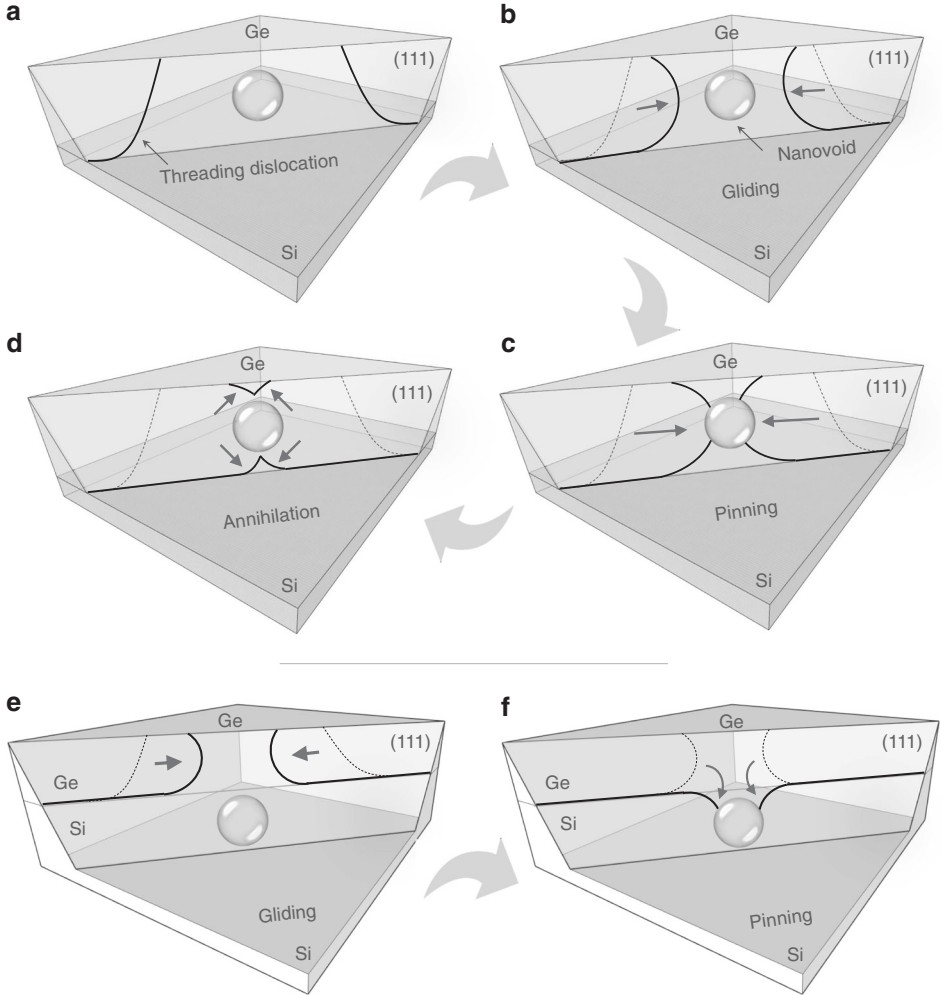

**Fig. 4** Processes involving dislocations-void rearrangements. By introducing nanovoids inside the Ge layer, the dislocations with opposite Burgers vectors move easily along the {111} glide planes during high-temperature annealing (**a**, **b**) and react with each other (**c**), allowing the threading components to disappear (**d**). When the voids are located at the Ge/Si interface (on the Si side), dislocations bend towards the porous area, instead of emerging at the Ge surface causing their elimination through a similar process at the nearest free surface (**e**, **f**)

confirmed and better understood using in situ real-time observations with TEM.

In order to study the effect of nanovoids formed in the Si substrate on reducing the TD density, low-magnification TEM micrographs from the Ge/Si reference, and the NVS are presented for comparison in Fig. 5a, b, respectively. In Fig. 5a, a high defect density region, extending from the heterointerface towards the surface may be observed for the Ge/Si reference. In Fig. 5b, we observe a dense dislocation network confined in a region of about 50 nm near the Ge/Si interface for the NVS sample and almost no observable defects within the Ge layer are shown. Fourier analysis from the Ge/Si interface has been performed (Supplementary Fig. 2), indicating that the epitaxial reconstruction of the Ge layer over the voided Si substrate has the same crystallographic orientation as the parent Si wafer. Figure 5c, d present enlarged images near to the Ge/Si interface for both substrates. For the Ge/ Si reference (Fig. 5c), it is seen that the only defects present in the Ge/Si heterostructure are misfit dislocations located at the Ge/Si interface. Several misfit dislocation cores marked by white arrows at the Ge/Si interface can be identified clearly. The type of these misfit dislocations was determined to be 60° dislocation and 90° full-edge dislocation from the high-resolution transmission electron microscopy (HRTEM) images by drawing a Burgers circuit around the dislocations. For the NVS (Fig. 5d), voids are observed as lighter areas. The amorphous zone appears in the void zone, probably due to the amorphisation of the void wall during the TEM sample milling or due the superposition of diffraction from the different crystalline material separated by voids. A rough interface at the atomic scale is observed. This indicates that the Ge/Si interface has undergone a strong perturbation during the electrochemical etching and annealing. If additional stress is applied, the interface becomes morphologically unstable and the roughness features could serve as nucleation sites for additional TD. To analyze the local strain distribution at the interface, we have used the geometric phase analysis (GPA) on the HRTEM images. Figure 5e, f present the $\varepsilon_{xx}$ component of the strain field ($x$-axis along the [100]) derived from Fig. 5c and the square from Fig. 5d, respectively. In the upper region (Ge), the strains are positive and tensile, while the lattice is compressed in the lower region (Si). On these images, the dislocation cores are easily located, as they correspond to the blue areas at the interface. To quantify the strain relaxation state, we calculate the average $\varepsilon_{xx}$ in the upper region (Ge) for both cases. The average value of $\varepsilon_{xx}$ in Ge/Si reference ($3.9 \pm 0.5\%$) is larger than the NVS ($3.2 \pm 0.5\%$) indicative of a better strain relaxation of the Ge layer in the NVS. These values are confirmed by calculating the distances from the transmitted beam spot to the paired separate spots in the selected area diffraction pattern

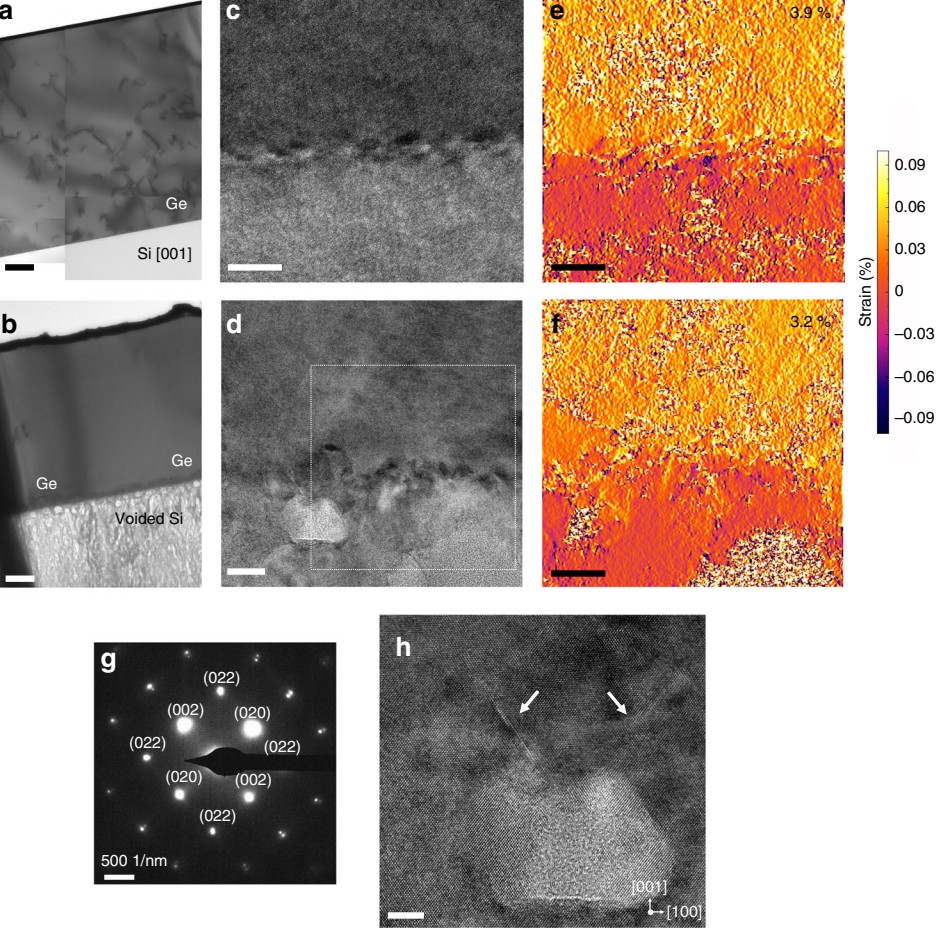

**Fig. 5** Dislocation/void interaction in the Ge/Si interface. **a** Low magnification bright field transmission electron microscopy (BF-TEM) images of the Ge/Si reference substrate and **b** the NVS with very few defects within the Ge layer. Scale bars 200 nm. High-resolution TEM images from the Ge/Si reference (**c**) and the NVS (**d**) show a dense dislocation network close to the Ge/Si interface. Scale bars of **c** and **d** are 10 and 20 nm, respectively. **e**, **f** Experimental strain components $\varepsilon_{xx}$ at the Ge/Si heterostructure interface for both substrates exhibit a better strain relaxation of the Ge layer in the case of the NVS. Scale bars 10 nm. **g** Selected area electron diffraction (SAED) patterns from the NVS show a monocrystalline quality of the Ge layer. **h** Atomic-resolution TEM image shows the annihilation of dislocation segments coming from the Ge/Si interface at the void located in the Si substrate. The arrows in **h** shows two dislocations bend towards the voided area, instead of emerging at the Ge surface. Scale bar 5 nm

(SADP), for both paired spots of (020) and (002) as shown in Fig. 5g. The lattice mismatch strain of the Ge film in the Ge/Si reference remains a little high, despite the plastic relaxation, which generates misfit dislocations. However, the Ge film in the NVS is almost fully relaxed, the strain relaxation is engineered elastically thanks to the voided Si substrate. Due to its porosity, the elastic modulus of Si is reduced and the substrate can be stretched, compressed, and deformed[74–76]. It is therefore expected to accommodate the mismatch of heterogeneous layers and to serve as a mechanically stretchable compliant substrate. In addition to this property, the nanovoids formed in the silicon substrate act as a free surface for inhibiting dislocation propagation. In fact, an enlarged image of the region near the Ge/Si interface by HRTEM, shown in Fig. 5h, clearly demonstrates two dislocation loops located within the heterointerface and annihilating at the voids located in the Si substrate, instead of emerging towards the Ge surface. The analog is also present in Si, where TDs originating from the interface bend downwards and move along the glide plane towards the voids in Si in order to minimize their length. Their elimination in pairs is also possible within these voids. A descriptive scheme of this phenomenon leading to the reduction of TD density when using a voided Si template is given in Fig. 4e, f. The attraction force is increased by

the fact that the presence of the nanovoids decreases the Si Young modulus, which might play a role in the apparent bending of TDs toward the porous Si layer. This is in good agreement with the analytical model proposed by Myers and Follstaedt[77].

In summary, for the NVS configuration, the voids located in the Si substrate offer elastic properties to accommodate the lattice mismatch. While, the voids located at the Ge film favor the recombination of TDs and their annihilation far from the surface. TD reduction is likely due to TD annihilation facilitated by nanovoids via the process shown in Fig. 4. Additionally, it was found that the existence of voids in a strained layer results in a considerable reduction in the stresses in the mid-region of the structure near the voids and consequently the dislocation density above the region[78]. This process was successfully used to produce wafer scale NVS on 4 in., as seen in Fig. 6a.

To quantify the reduction in TD density with lower magnification images, etch-pit density (EPD) analyses were carried out. The NVS was immersed in a solution of two volumetric parts 49 wt% HF and 1 part 0.1 M $K_2Cr_2O_7$, to selectively etch the mixed and screw dislocations in the Ge layer. The Supplementary Fig. 3 shows the top view microscope images revealing a number of dark pits at the center. Figure 6b–d show SEM images of the EPD taken from the edge of the NVS with an

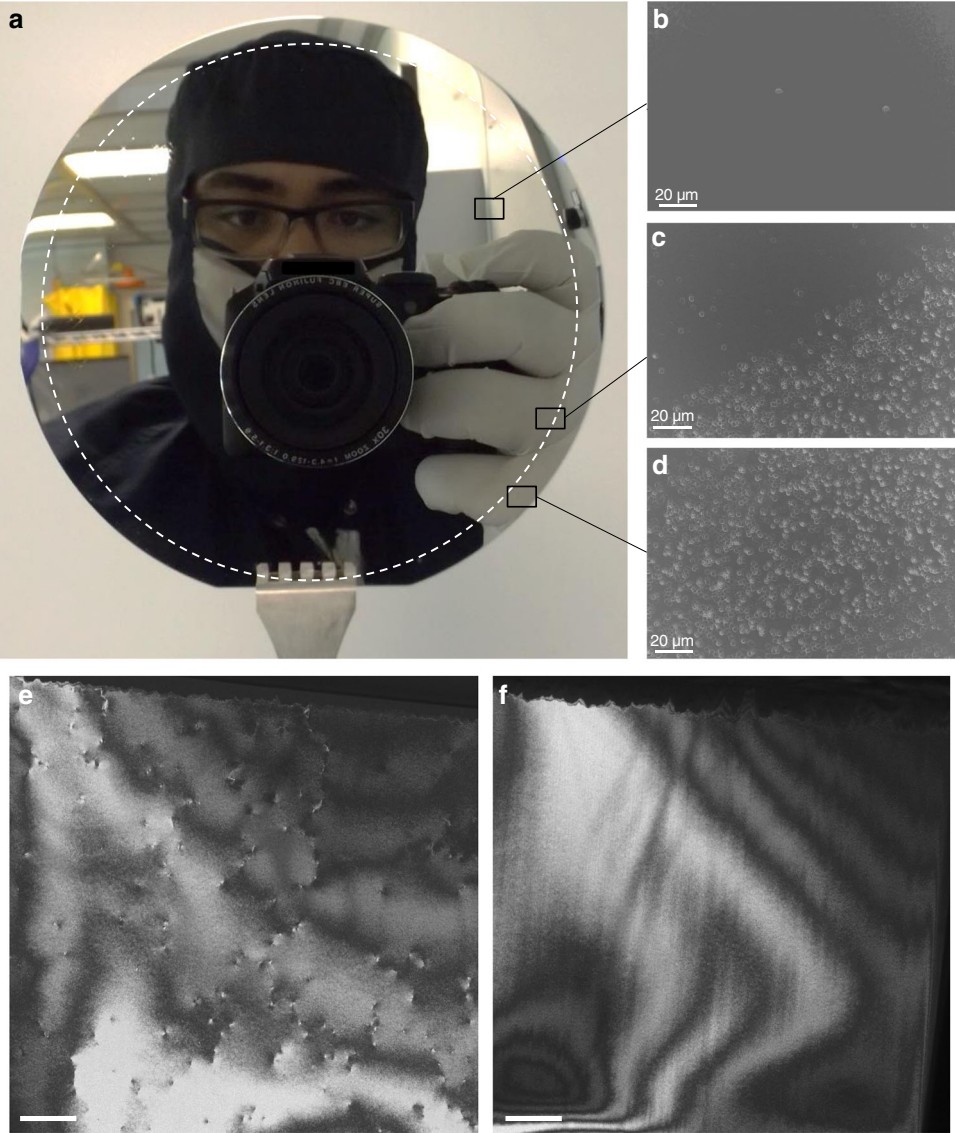

**Fig. 6** Evaluation of threading dislocation densities. **a** Picture of the nanovoid-based Ge/Si virtual substrate (NVS) produced on a 4 in. wafer. Etch-pits densities taken from different regions of the NVS providing a lower limit for the TD density of ~$10^4$ cm$^{-2}$ for 1.5 μm-thick Ge layer (**b**), from the edge of the NVS with an untreated area (**c**) and from an untreated area with a TD density of ~$10^7$ cm$^{-2}$ (**d**). Plan-view TEM micrographs marking the emergence sites of dislocations on Ge films on Si taking from **e** untreated Ge/Si sample and **f** the NVS sample. Scale bars 500 nm. These observations confirm the TD density reduction by introducing nanovoids inside the Ge/Si substrate

untreated area, and from different regions of the sample. From the pit counts, the EPD is clearly lower in the voided area compared with the untreated area, which confirms that the dislocations were effectively annihilated in the case of the NVS sample.

The TD density can also be determined from the plan-view TEM images by estimating the dislocations in a given area at a number of zones across the entire samples[79]. Figure 6e, f shows two-TEM plane-view images for an untreated Ge/Si substrate and a NVS, respectively. The contrast in the images is dominated by thickness fringes and bend contours, which are inevitable in plan-view crystalline samples[80,81]. The dark spots show dislocation strain-induced bending of the thinned film. The estimated TD density from Fig. 6e is $8.5 \pm 0.5 \times 10^8$ cm$^{-2}$, while no threading dislocation is found in most of the areas of the NVS as shown in Fig. 6f (within statistical limits). While the fact that TEM images on an area of 15 μm$^2$ do not show any dislocation does not constitute a quantitative measurement, it does imply that the actual TD density must be at least an order of magnitude below a

value of ~$10^6$ cm$^{-2}$, which would correspond to an average distance of about 10 μm between dislocations. Considering the high correlation between the TD density measured by plan-view TEM and EPD[79], we argue that the ~$10^4$ cm$^{-2}$ figure for the TD density as measured by EPD is reasonably representative and should be comparable with other results from the literature measured in the same manner. By combining different interactions between dislocations and voids, located either in the Ge layer or in the Si substrate, the averaged TD density is reduced significantly from ~$10^8$ cm$^{-2}$ to a lower limit of ~$10^4$ cm$^{-2}$ for 1.5 μm-thick Ge layers, which is considered very low for such a thin epitaxial layer[32,33,82].

**Room temperature optical emission.** As the TD density is significantly reduced, one would expect that the NVS could be an ideal starting point for the growth of high-quality III–V alloys and devices. To demonstrate this, an epitaxial growth of 300 nm-thick GaAs layers was carried out on three substrates for

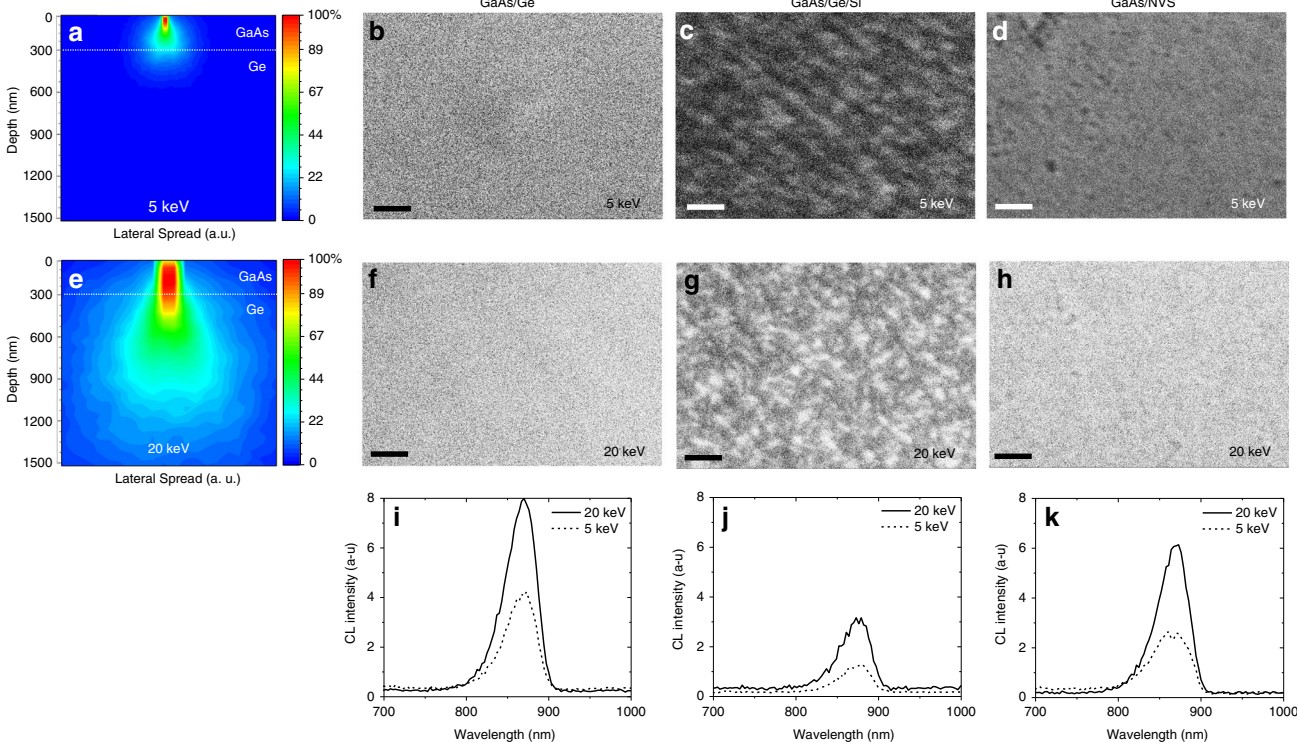

**Fig. 7** Effect of reducing dislocations on optical properties. Monte-Carlo simulations using CASINO software of the electron interaction volume, and planar cathodoluminescence (CL) micrographs of a 300 nm-thick GaAs layer grown on bulk Ge substrate, Ge/Si substrate, and the NVS at 5 keV (**a**–**d**) and 20 keV (**e**–**h**), respectively. The color scale bar in **a**, **e** shows the electron concentration. The CL micrographs in **b**–**h** show the emission and the recombination behavior at the surface (5 keV) and in the bulk (20 keV) of GaAs on each substrate. Scale bars 3 μm. Room temperature cathodoluminescence spectra recorded from **i** GaAs/Ge, **j** GaAs/Ge/Si with a TD density of ~$10^7$ cm$^{-2}$ and **k** GaAs/NVS with a TD density of ~$10^4$ cm$^{-2}$, at 5 and 20 keV. CL measurements confirm that the dislocations were effectively blocked from propagating using nanovoids, which is reflected by an enhancement of the emission efficiency of GaAs grown upon the NVS

comparison: bulk Ge, commercial Ge/Si with a TD density of ~$10^8$ cm$^{-2}$ and the NVS with a TD density of ~$10^4$ cm$^{-2}$. Light emission efficiency and optical emission spectra from these substrates were measured by CL.

First, the use of surface preparation before the growth always led to a noticeable reduction of the surface roughness. The calibrations of the RMS for various chemical–mechanical planarization (CMP) durations were carried out on a rough Ge/Si substrate using a mixture of commercial CMP slurries, DI water dilution, and 1 wt% $H_2O_2$ (Supplementary Fig. 4). Using this process, the initial RMS of the NVS (2 nm for a large window size of $20 \times 20$ μm$^2$) was reduced to 0.5 nm, to fulfill the requirements for the epitaxy of III–V alloys. The polished thickness of the NVS is around 40 nm.

Optical emission properties of the GaAs films were investigated under electron irradiation using CL measurements. The energy-loss and CL generation profiles simulated for a 300 nm-thick epilayer of GaAs (density 5.32 g/cm$^2$) on Ge (density 5.32 g/cm$^2$), for monoenergetic beams of 5, 10, and 20 keV are given in Supplementary Fig. 5. These parameters were calculated using the Monte Carlo simulation CASINO[83] package in order to optimize the CL emission depth profile of GaAs films and compare the recombination behavior at the surface and in the bulk for each substrate.

CASINO simulation of the electron interaction volume of a 300 nm-thick GaAs film grown on Ge substrate is presented in Fig. 7a, e. The color scale indicates the irradiated electron energy concentration. The thickness of GaAs used in this work has been overlaid in the diagram to facilitate comparison. Simulations show that interactions occurring near the surface could be

revealed using 5 keV e-beam and across the entire layer of GaAs using 20 keV e-beam.

CL spectra from all samples exhibit luminescence in the near-infrared region with a maximum intensity at 870 nm as shown in Fig. 7i–k. The peak is correlated to the interband recombination process of excited charge carriers across the direct bandgap of GaAs (1.43 eV) at room temperature[84]. At the surface (5 keV e-beam), the recombination process is less radiative for all samples due to a high density of surface states (Fig. 7b–d).

For GaAs grown on bulk Ge (Fig. 7i), the detected CL radiation is higher than that of GaAs grown on Ge/Si (Fig. 7j) and NVS substrates (Fig. 7k). This is to be expected, since GaAs and Ge have similar lattice constants and a good quality growth is easily achievable in these conditions. GaAs layers were deposited using a recipe and surface preparation that are optimized to minimize antiphase boundary (APB) formation, which can affect the luminescence. Our material does not show any such APB under SEM observation. CL maps demonstrate the uniformity of the grown GaAs top layer, reflecting the absence of nonradiative recombination centers and charged defects. No additional peaks associated with Ge acceptor or donor levels have been observed in our CL spectra, indicating that any diffusion of the Ge atoms from the template into the GaAs epilayer is negligible[85,86]. The CL intensity of GaAs on Ge/Si substrate is very low to compare with GaAs on bulk Ge. The correlation between CL spectra and CL micrographs shows that this attenuation is due to the electrical activity of defects, which reveals dark regions, where the recombination is non-radiative. These defects are mainly the TD present in the GaAs layer, which propagate from the mismatched Ge layer on Si. For the NVS, we observe a significant recovery of

the CL intensity, when compared with the GaAs/Ge/Si sample. The comparison of CL micrographs between the two samples shows the disappearance of dark regions, which corresponds to a low threading dislocation density inside the GaAs layer, as shown in Fig. 7h. These results confirm that the dislocations were effectively annihilated from the epilayer surface, which is reflected by an enhancement of the emission efficiency for GaAs grown upon the NVS.

## Discussion

Low TD density in mismatched Ge-on-Si substrate is produced by an innovative approach, which consists in trapping and annihilating dislocations by self-assembling nanovoids close to the Ge/Si interface. The nanovoids are formed in the Ge layer as well as in the Si substrate by electrochemical porosification followed by thermal annealing. The results show that PGe is selectively formed through etching of the threading dislocation cores with higher etch rate than for the defect-free regions, up to full etching of misfit dislocations. We have demonstrated by TEM analysis different effects of either of the free wedge surfaces on dislocation, particularly, threading arm pinning by nanovoids. Possible mechanisms responsible for TD reduction that lead to either annihilation or fusion have been discussed. The results suggest that introducing nanovoids favors recombination of TDs by increasing the characteristic interaction distance between neighboring dislocations. In fact, the creation of voids could facilitate interactions between dislocations, enabling the dislocation network to change its connectivity in a way, which facilitates the subsequent annihilation of TD segments. In addition, the voids formed in the silicon substrate could potentially capture and thereby combine many more TDs. The TDs bend towards the voids in Si in order to minimize their length, causing their recombination and elimination at the nearest free surface thus leading to the creation of an almost defect-free Ge layer on Si. CL measurements indicate a strong enhancement of emission efficiency in GaAs grown on this Ge/Si virtual substrate (with a lower limit for a TD density of $\sim10^4$ cm$^{-2}$) to compare with a commercial Ge/Si (with a TD density of $\sim10^8$ cm$^{-2}$) confirming that TDs were effectively blocked from propagating, thanks to the nanovoids. The use of such a simple, inexpensive process through electrochemical etching and thermal annealing could be crucial in cutting manufacturing costs and enabling market penetration of III–V on Si-based devices.

## Methods

**Ge/Si porosification**. A p+-type Ge layer of 1.5 μm was grown on nominal B-doped Si (001) wafers (4 in. diameter) using ultrahigh vacuum chemical vapor deposition (UHV-CVD), with an initial TD density of $\sim10^7$ cm$^{-2}$. Mesoporous Ge structures were fabricated by BEE in a two-electrode electrochemical cell with a platinum wire counter electrode using an electrolyte consisting of 5:1 HF (49%) and anhydrous ethanol. Anodic and cathodic currents with densities of ±4 mA/cm$^2$ were applied alternately with pulse durations of 1 s during 35 min to fabricate PGe/Si sample.

**Thermal annealing**. Annealing of mesoporous Ge/Si structures was performed in a forming gas (N$_2$:H$_2$ 90:10) ambient using a J.I.P. ELEC JetFirst rapid thermal annealing (RTA) system with a ramp rate of 25 °C/s. The duration was fixed at 10 min and the annealing was performed between 300 and 600 °C to obtain the voids in the Ge layer as well as the Si substrate.

**Epitaxial growth**. GaAs growth was carried out in a VG Semicon VG90H CBE reactor. A 300 nm layer of GaAs was deposited using triethylgallium (TEGa) and cracked arsine (AsH$_3$) as the Groups III and V sources, respectively. Prior to growth, the Ge substrates were heated at a temperature of 650 °C in the growth chamber (under vacuum) for 10 min in order to allow for complete oxide desorption. The growth run was carried out at 550 °C.

**Structural characterization via TEM**. The samples were observed in a scanning/transmission electron microscope (S/TEM) in the high-angle annular dark field (HAADF) mode using a Titan Themis microscope operated at 200 kV and equipped with a CEOS probe corrector and a Ceta 16M camera from FEI. The sample was prepared for S/TEM using focused ion beam (FIB) thinning and ion milling. 100 nm-thick carbon was deposited before the FIB step in order to protect the surface. Elemental distribution analysis was carried out using the Gatan digital micrograph (DM) and energy dispersive X-ray spectroscopy (EDX) combined with STEM. For EPD, the etchant was a mixture of two volumetric parts 49 wt% HF and 1 part 0.1 M K$_2$Cr$_2$O$_7$. Etch pits were counted on the top surface by examining SEM images and averaged over several samples. Additional nanoscale structural information has been obtained by means of μ-Raman spectroscopy at room temperature using an excitation wavelength of 532 nm and an excitation power of 1 mW/cm$^2$.

**Cathodoluminescence**. The CL spectra and images are acquired at room temperature, in the same SEM setup as the one used for imaging the layers. Our CL system, in association with a spectrometer, which allows for monochromatic CL (GATAN MonoCL2) imaging, as well as the acquisition of CL spectra on localized spots of a sample with a spectral resolution of 0.5 nm. The accelerating voltages used in the CL characterization are 5 and 20 keV.

## Data availability

All relevant data are available from the authors upon request.

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

## Acknowledgements

The authors would like to thank SUNLAB group at the University of Ottawa, K. Schulte from NREL and M. Rondeau for fruitful discussions, D. Troadec for his help in TEM sample preparation, G. Beaudin for 3D illustrations, E. Paradis and S. Ecoffey for their help, NSERC strategic project grant program; and RQMP and FQRNT for financial support.

## Author contributions

Y.A.B. performed the experiments, analyzed the data, and wrote the manuscript. M.M. developed Ge growth, A.S. prepared TEM samples, Y.A.B. and N.B. carried out GPA analyses, characterization of HR-TEM was performed by Y.A.B. and G.P. CL was measured by Y.A.B. The manuscript was revised by all authors. The project was planned, directed, and supervised by D.D., A.B. and R.A.

## Competing interests

The authors declare no competing interests.
