## [Peer Review File · Nature Communications]

Reviewers' comments:

Reviewer #1 (Remarks to the Author):

The manuscript presents a novel methodology for reducing threading dislocations (TDs) in Si/Ge substrates. The paper is significant because the problem addressed by the proposed methodology, namely the accumulation of threading dislocations due to lattice mismatch of epitaxial and heteroepitaxial films, is a primary impediment to improved device performance and bandgap design. The multistep methodology is clearly presented: (1) epitaxial growth of Ge on Si with commensurate threading/misfit dislocations, (2) electrochemical etching that preferentially etches threading dislocations, and (3) thermally-induced reorganization (annealing) of the etched regions. This process creates what the authors refer to as a Nanovoid-based Ge/Si Virtual Substrate (NVS). Etch pit density results showed a significant reduction of threading dislocations. Utilizing NVS as a substrate for GaAs showed significant improved in luminescence over conventional GaAs/SiGe constructs. Overall the conclusions drawn by the authors are supported by the results presented and the reviewer recommends after addressing the following minor concerns:

1) The authors claim that their method provides an inexpensive methodology for reduction of TDs in heteroepitaxial films. It is not immediately obvious that this claim is supported in the manuscript. Please provide a discussion of why this approach is more cost effective than the competing alternatives described in the 1st paragraph.

2) A variety of phenomena are described for why this process reduces threading dislocations. However, an important mechanism has been overlooked, namely, the presence of stress inhomogeneity. The existence of a void in a strained layer creates regions of stress inhomogeneity in the film(s); additionally, the misfit dislocation structure provides a strongly inhomogeneous stress field. Previous researchers have shown that the presence of stress inhomogeneity biases threads to interact and annihilate (Schwarz and Tu 2009, Fertig III and Baker 2010), which will serve to reduce threading dislocation density in films.

3) The authors note that after the annealing step, the interface between Si and Ge is rough. It would seem that these roughness features could serve as nucleation sites for additional threading dislocations if additional stresses were applied, such as might occur under additional changes in temperature in situ or during processing of the NVS substrate for use. A discussion of this issue is warranted.

4) Related to comment (3). The modulus of Ge is less than that of Si. As such, the Si/Ge boundary will tend to repel any dislocations (via image forces) that approach the boundary. By making the Si near the boundary porous and reducing the modulus, this repulsion will be mitigate. In fact, it might be reversed, depending on if the modulus in the vicinity is lower than the Ge, such that any additional dislocations created are attracted to the interface.

5) The luminescence results are primarily a comparison between NVS and Ge/Si as a substrate. But both are compared with pure Ge as a substrate with no discussion as to why the best luminescence is achieved with pure Ge.

References:

Fertig III, R. S. and S. P. Baker (2010). "Dislocation dynamics simulations of dislocation interactions and stresses in thin films." *Acta Materialia* 58(15): 5206-5218.

Schwarz, K. W. and Y. H. Tu (2009). "Dislocation-interaction-based model of strained-layer relaxation." *Journal of Applied Physics* 106(8).

Reviewer #2 (Remarks to the Author):

This paper describes a method for preferentially etching the material around dislocation cores in a Ge/Si epilayer, then annealing the material. This 2-step process enables two things: (1) movement and re-arrangement of dislocations through the porosified material and (2) re-solidification of the

Ge epilayer.

This method definitely merits exploration and publication of the results, and the authors deserve credit for their work. However, there are two major problems with the paper in its current form, and it should not be published without addressing them.

1) The claimed threading dislocation density (TDD) of $1 \times 10^{-4} \text{ cm}^{-2}$ is remarkably low, and such a claim should be bolstered with a direct measurement of the TDD by, for example, plan-view TEM. The concern would be that etch-pits may only be revealing a fraction of the threading dislocations (TDs), and thereby providing only a lower-limit for TDD. For example, etch pits may only be revealing bunches of TDs in close spatial proximity, and not revealing isolated TDs. Etch pits may ultimately be the correct way to assess a very low TDD, but a one-to-one correspondence between TDs and etch pits needs to be solidly established for the reader.

CL data is also offered, but this is also an indirect measure of TDD, and it isn't clear if CL results can discriminate between (for example) 1×10^6 and $1 \times 10^4 \text{ cm}^{-2}$.

2) If the TDD of $1 \times 10^{-4} \text{ cm}^{-2}$ can be conclusively proven, then this experimental result is strong enough to merit publication with only a few comments about possible TD-removal mechanisms. Some of the discussion in the current version is insightful and should be retained. In particular, the creation of voids could facilitate interactions between dislocations, enabling the dislocation network to change its connectivity in a way which facilitates the subsequent annihilation of TD segments. The etching of the interfacial misfit dislocation (MD) network may also play a crucial role, so this should also be mentioned as a possible enabling step unique to this process.

However, the paper would be strengthened if much of the speculation beyond this were simply removed, because the current version of the text perpetuates many misconceptions about dislocations, and does not address how the TDD at the TOP of the Ge epilayer is reduced.

Perhaps the best example is Figure 1, which sets the stage for confusion throughout the remainder of the paper. The "before" image shows two dislocation half-loops, but the remainder of the TDs appear to end abruptly at the Ge/Si interface. To provide a solid basis for discussion, all TDs should be either connected pairwise with MDs to form half-loops, or continue as TDs into the Si substrate. Where MDs are concealed within the 3D image, perhaps dashed lines could be used.

In the middle panel of Figure 1, all dislocations have been omitted, so absolutely no information about what the porosification might be doing to the TD network is conveyed.

In the "after" panel of Figure 1, isolated dislocations appear to terminate at voids, and this is topologically impossible. Wherever there appears to be a single dislocation entering a void, there must be at least one other dislocation entering or exiting it. To convey meaningful information, these additional dislocation segments should be included, perhaps with dashed lines if they travel through the middle of the 3D rendering.

Perhaps most importantly, though, there is no illustration for how the TDs in the "before" panel have either climbed, glided or combined to reduce the TDD at the top of the Ge layer. All of the problems in the text follow from this.

Therefore, my recommendation is to (A) replace Figure 1 with a set of diagrams which correctly, completely and sequentially illustrate a process by which the TDD at the top of the Ge layer is reduced, then (B) rewrite all subsequent associated text describing this process. If this can't be done, then it would be best to provide instead a brief discussion of possible dislocation annihilation mechanisms, along with an acknowledgement that the full process is not understood at this point.

This same advice applies to all of the subsequent statements about dislocations in the paper.

For example (lines 274-276) "In Si, TDs bend into the glide plane towards the voids in Si onto the glide plane in order to minimize their length, causing their elimination at the nearest free surface." It should be possible to illustrate this sentence with a topologically correct sequence of diagrams, and doing this should ultimately help explain how the TDD at the top of the Ge layer is reduced. If this can't be done, consider omitting this sentence, and many other similar sentences.

Finally, a quick comment on the title. There seems to be a mismatch between prominence of "light-emitting III-V on Si" in the title and the introduction of the paper which mostly discusses III-V solar cells (and not LEDs or lasers).

Reviewer #3 (Remarks to the Author):

Uprooting defects to enable light-emitting III-V on Si

Y. A. Bioud, A. Boucherif, A. Soltani, N. Braidy, D. Drouin, R. Arès

This paper reports on a novel method that allows reducing the dislocation density in lattice-mismatched hetero-epitaxy semiconductor systems. The ingenuity of the method resides on the use of a nanovoid lattice to block dislocations and thereby favor their annihilation. It is very interesting and the method was thoroughly checked by transmission electron microscopy techniques.

Structure of the paper is good. The writing is relatively good, but deserves improvement here and there. The use of the tense of the verbs is at times awkward. For example, in the introduction it is said that '... have been discussed.' for a topic that is discussed in the present paper. I suggest to use the present tense instead. The use of the verb 'show' should be checked (use instead e.g. 'exhibit', 'present'). Figures and graphs are clear.

There are however a number of issues the authors should address before the paper can be accepted for publication.

The part explaining the various dislocation mechanisms at play in their system is confused if not weak:

'Different phenomena may come into play in the dislocation pinning mechanism, which are all promoted by high temperature achieved during annealing including; gliding,[50] climbing processes, [51,52] the void size effect[53,54] and inertial effects.'

1) This is a vague statement, what is meant by 'promoted' in this context? does it mean 'enhanced'? In this case one cannot state it as a generality: , i.e. temperature does not 'promote' all the different phenomena described here. For instance, the obstacle strength will actually decrease when temperature increases (for voids: see Fig. 8 in Journal of Nuclear Materials 382 (2008) 147–153 and for a comparison between different types of obstacles: see Fig. 6 in Acta Materialia 64 (2014) 24–32).

2) 'void size effect': This doesn't grasp all elements needed to rationalize dislocation-void interaction: actually it is about the size and the number density of voids (this can be summarized by the distance between voids, surface to surface).

3) 'inertial effects': This concerns the detailed dynamics of the dislocation-defect interaction. In this study these effects can neglected, as the interest here is to consider the dislocation configuration at equilibrium. I wouldn't even mention inertial effects.

4) 'For large voids that are sufficiently spaced out, dislocation can bypass the obstacle via the Orowan mechanism.'

This is wrong, this doesn't occur for voids. 'Orowan mechanism means' the formation of a dislocation loop around the obstacle. However, in this case (a void) the segment of the dislocation that meets the void will disappear in the void (making a step in the internal surface; this step has

the amplitude of the Burgers vector of the dislocation). Actually, a dislocation is attracted by the internal surface of the void: as this surface is a free surface, it induces image forces on the dislocation that attract the dislocation. The dislocation therefore will not stop before the surface of the void and thus cannot form an Orowan loop around it.

This absorption is well explained in the paper of Crone 2015 cited by the authors (J. C. Crone, L. B. Munday, J. Knap, *Acta Mater*, 2015, 101, 40–47).

5) Besides the direct interaction of the dislocation with the void, there is also the interaction of the dislocation with the stress field induced by the void in its surroundings (see *Philosophical Magazine* 90 (7–8) (2010) 1075–1100). This is missed by the authors.

Crystallographic index notation: for a direction, it should be $[\]$ (particular) or $\langle \rangle$ (general), for a plane it should be $()$ (particular) or $\{ \}$ (general). Page 10: '(x-axis along the $[100]$)'; authors have chosen a specific $\langle 100 \rangle$ -type direction. From this choice, the orientations following in the text should be noted as:

'for both paired spots of 020 and 002' -> 'for both paired spots of (020) and (002)'. I suggest to change '000 spot' to 'transmitted beam spot'.

Page 9: 'Due to the attractive force created by the population of smaller voids, the approach distance between two TDs with different Burgers reaches a low value in which the TDs begin to glide together and annihilate or fuse. These mechanisms give a reasonable explanation for the dislocations reduction by introducing nanovoids in heteroepitaxial diamond films, in which, the TDs have an interaction radius higher than the interaction radius in the conventional structure without voids, which in turn increases their recombination probability.'

-> 'TDs have an interaction radius higher' This conclusion is dumped from high: it is not justified, it is not quantified (how many nanometers? calculated on the base of what?). The simplest conclusion one can draw in this case is that one can assume the interaction radius is the same but the dislocation density being higher the distance between them is smaller, so the probability of annihilation increases.

Page 10: 'Due to its porosity, the elastic modulus of Si is reduced and the substrate can be stretched, compressed, and deformed. It is therefore expected to accommodate the mismatch of heterogeneous layers and to serve as a mechanically stretchable compliant substrate.'

Did the author checked the mechanical integrity? Authors should really check the mechanical integrity of the resulting device. Such an array of nanovoids introduces a fragilized area that may lead to the premature failure of the whole device during transport and handling. Authors should add a comment in the text about this point.

Responses to Reviewers:

Please find attached a revised version of our manuscript, which we would like to resubmit for publication in *Nature Communications*. The reviewers have clearly made an effort to read the paper carefully and have raised several excellent and insightful points, which have helped us to improve its quality.

In the following pages are our point-by-point responses to each of the reviewers' comments. We will refer each of the comments as, for example, 'R1.2' meaning reviewer #1's second comment.

All modifications are highlighted with text in yellow background.

Reviewer #1

The manuscript presents a novel methodology for reducing threading dislocations (TDs) in Si/Ge substrates. The paper is significant because the problem addressed by the proposed methodology, namely the accumulation of threading dislocations due to lattice mismatch of epitaxial and heteroepitaxial films, is a primary impediment to improved device performance and bandgap design. The multistep methodology is clearly presented: (1) epitaxial growth of Ge on Si with commensurate threading/misfit dislocations, (2) electrochemical etching that preferentially etches threading dislocations, and (3) thermally-induced reorganization (annealing) of the etched regions. This process creates what the authors refer to as a Nanovoid-based Ge/Si Virtual Substrate (NVS). Etch pit density results showed a significant reduction of threading dislocations. Utilizing NVS as a substrate for GaAs showed significant improved in luminescence over conventional GaAs/SiGe constructs. Overall, the conclusions drawn by the authors are supported by the results presented and the reviewer recommends after addressing the following minor concerns:

R1.1 The authors claim that their method provides an inexpensive methodology for reduction of TDs in heteroepitaxial films. It is not immediately obvious that this claim is supported in the manuscript. Please provide a discussion of why this approach is more cost effective than the competing alternatives described in the 1st paragraph.

Thank you for the comment.

Multiple approaches have been studied to reduce the TDD of the epitaxial Ge films. Although technical reports have been published, the improvement of device performance after these processes is still very limited in regions with high dislocation densities ($\sim 10^6 \text{ cm}^{-2}$). Furthermore, many of these techniques are limited to small-scale processes and require the use of complex processing technologies, which involves significant technological costs.

As examples:

- Graded SiGe buffers are one way to gradually reduce the lattice mismatch and therefore the TDD. However, a 10 μm thick buffer layer and chemical mechanical polishing are required, which results in high manufacturing costs [1].
- Defect blocking through substrate patterning is another method to reduce the TDD, which involves expensive and low-throughput nanometer level patterning.

This paper describes an approach involving only a basic 2-step process (electrochemical etching and thermal annealing) that enables two things: (1) movement and re-arrangement of dislocations through the porosified material and (2) re-solidification of the Ge epilayer.

Advantages of this technique include low cost, large surface area, and compatibility with microfabrication facilities and other processing steps. The key cost-reducing aspects of this process flow are the intrinsic compatibility of each process step with large area semiconductor mass production. Both electrochemical porosification and thermal annealing are standard, scalable techniques that are already well integrated within the industry. Our process requires no rare or special material or furniture, nor does it take long processing times to perform. Current production tools are available and well established for the process. As an indicator, a detailed study of the cost items in the integration of III-V multifunction solar cell on the nanovoid-based Ge/Si virtual substrate are listed in Table I (note that we made rough estimations in some cases that may not be particularly trustworthy.). The price of this 2-step process in high throughput systems is estimated by J. Scott Ward *et al.* to be ~ \$1 for 8" Si wafer [2].

It is our intent in this manuscript to focus mainly on TD-removal mechanisms and the luminescence properties that may be achievable for certain levels of material quality rather than make assumptions about the eventual production cost of this device relative to the market. We have however emphasized this point by adding the following sentence:

“The approach proposed herein involves only a basic, industry standard 2-step process (electrochemical etching and thermal annealing); which we expect will likely have a small impact on the overall processing costs of a device. This process enables two things: (i) movement and rearrangement of dislocations through the porosified material and (ii) re-solidification of the Ge epilayer. Advantages of this technique include low cost, large surface area, and compatibility with microfabrication facilities and other processing steps. The key cost-reducing aspects of this process flow are the intrinsic compatibility of each process step with large area semiconductor mass production. Both electrochemical porosification and thermal annealing are standard, scalable techniques that are already well integrated within the industry. Our process requires no rare or special material or treatment, nor does it take long processing times to perform. Current production tools for this process are also readily available and well established.”

3J on a Nanovoid-based Ge/Si Virtual Substrate				
		Cell area (cm ²)	237	
	Details	Value (\$/m ²)	Value (\$/cell)	Source
Substrate	8" Si wafer 160 um thick 237 cm ² pseudo square cell		\$1,00	NREL - A Manufacturing Cost Analysis Relevant to Single- and Dual Junction Photovoltaic Cells Fabricated with III-Vs and III-Vs Grown on Czochralski Silicon
Ge epi, porosification, & annealing			\$1,00	Ward, Progress in Photovoltaics, 2016
Ge overgrowth (5 um)			\$1,00	
Standard 3J Material costs	Lattice matched 3J cell	\$1 900,00	\$45,03	
Labor + Utilities + Maintenance + Depreciation		\$1 000,00	\$23,70	NREL - A bottom-up cost analysis of a high concentration PV module
Overhead + margin to meet MSP		\$928,00	\$21,99	
Total (cell)			\$93,72	
		Module area (m ²)	1,98	
	Details	Value (\$/m ²)	Value (\$/module)	Source
Module extras per area	1.98 m ² Si module	\$31,00	\$61,38	Goodrich, Solar Energy Materials & Solar Cells, 2013
Module extras flat rate			\$29,00	
Total (module)			\$90,38	
		# cells per modul	72	
Total (cell + module)			\$6 838,48	
Cell efficiency (%)			33	
Incident power (W/m ²)	1 sun standard		1000	
Module power (W)			563,112	
Price of power (\$/W)			\$12,14	

Table 1 Cost estimate for Ge porosification/annealing step and the price of power (\$/W) for III-V MJSC on a Nanovoid-based Ge/Si Virtual Substrate.

R1.2 A variety of phenomena are described for why this process reduces threading dislocations. However, an important mechanism has been overlooked, namely, the presence of stress inhomogeneity. The existence of a void in a strained layer creates regions of stress inhomogeneity in the film(s); additionally, the misfit dislocation structure provides a strongly inhomogeneous stress field. Previous researchers have shown that the presence of stress inhomogeneity biases threads to interact and annihilate (Schwarz and Tu 2009, Fertig III and Baker 2010), which will serve to reduce threading dislocation density in films.

Thanks for the interesting references. It seems that the reviewer 3 also raises the question regarding “the interaction of the dislocation with the stress field induced by the void”. We added a paragraph in this context (please see **R3.2.d**).

In addition, we have emphasized this point by adding another sentence:

“it was found that the existence of voids in a strained layer results in considerable reduction in the stresses in the mid-region of the structure near the voids and consequently the dislocation density above the region [3].”

R1.3 The authors note that after the annealing step, the interface between Si and Ge is rough. It would seem that these roughness features could serve as nucleation sites for additional threading dislocations if additional stresses were applied, such as might occur under additional changes in temperature in situ or during processing of the NVS substrate for use. A discussion of this issue is warranted.

Thanks for the comment. Figure S3a-c from the Supporting Information shows an EDX mapping of the NVS revealing an unclear, fuzzy, Ge/Si interface, probably due to a mixed reorganization of the porous Si and Ge atoms at the interface during thermal annealing. However, the extent of this interface represents intermixing on only a few monolayers. We agree with the reviewer that a rough interface could generate nucleation sites for additional threading dislocations. However, we argue that while the distributed interface. Might have a corresponding stress field, it is unlikely to be larger than the one that was present after the epitaxial deposition of Ge on Si with an abrupt interface. New dislocations might be nucleated at some points during annealing, but others will be annihilated so the overall number remains at most constant. We currently do not have the means to quantify its impact, as it is at an atomic scale and we do not have a “before and after” image of the same region that could answer the question. We have concluded that the final dislocation density can be reduced by using this process, even if the origin of single dislocation could be transferred between different points in the interface. We have highlighted this point by adding the following sentence:

“If additional stress is applied, the interface becomes morphologically unstable and the roughness features could serve as nucleation sites for additional threading dislocations.”

R1.4 Related to comment (3). The modulus of Ge is less than that of Si. As such, the Si/Ge boundary will tend to repel any dislocations (via image forces) that approach the boundary. By making the Si near the boundary porous and reducing the modulus, this repulsion will be mitigate. In fact, it might be reversed, depending on if the modulus in the vicinity is lower than the Ge, such that any additional dislocations created are attracted to the interface.

That is a pertinent question. Yes, it is possible that the porous Si substrate becomes more attractive for dislocations by reducing its Young's modulus (depending on (i) if the modulus of porous Si is lower than the Ge and (ii) if the Ge layer is not porous). In fact, porous Si is characterized by a Young's modulus that can be adjusted in the range of 120–20 GPa by varying the porosity [4]. Consequently, this driving force might play a role in the apparent bending of TDs toward the porous Si layer and could certainly stimulate the process of trapping dislocations at the porous interface. It is important to make a rigorous study to determine what porosity of Si and what thickness of Ge necessary to attract dislocations downwards instead of emerging at the Ge surface. At this stage, we have emphasized this effect by adding the following sentence:

“The attraction force is increased by the fact that the presence of the nanovoids decreases Si Young modulus, which might play a role in the apparent bending of TDs toward the porous Si layer. This is in good agreement with the analytical model proposed by Myers and Follstaedt [5].”

R1.5 The luminescence results are primarily a comparison between NVS and Ge/Si as a substrate. But both are compared with pure Ge as a substrate with no discussion as to why the best luminescence is achieved with pure Ge.

Thanks for the comment. Indeed, this is an important point. In the manuscript, we have clarified this point by adding the following paragraph:

“For GaAs grown on bulk Ge, the detected cathodoluminescence radiation is higher than that of GaAs grown on Ge/Si and NVS substrates. This is to be expected since GaAs and Ge have similar lattice constants and a good quality growth is easily achievable in these conditions. GaAs layers were deposited using a recipe and surface preparation that are optimized to minimize antiphase boundary (APB) formation, which can affect the luminescence. Our material does not show any such APB under SEM observation. CL maps demonstrate the uniformity of the grown GaAs top layer, reflecting the absence of nonradiative recombination centres and charged defects. No additional peaks associated with Ge acceptor or donor levels have been observed in our CL spectra, indicating that any diffusion of the Ge atoms from the template into the GaAs epilayer is negligible [6], [7].”

Reviewer #2

This paper describes a method for preferentially etching the material around dislocation cores in a Ge/Si epilayer, then annealing the material. This 2-step process enables two things: (1) movement and re-arrangement of dislocations through the porosified material and (2) re-solidification of the Ge epilayer. This method definitely merits exploration and publication of the results, and the authors deserve credit for their work. However, there are two major problems with the paper in its current form, and it should not be published without addressing them.

R2.1 The claimed threading dislocation density (TDD) of $1e-4 \text{ cm}^{-2}$ is remarkably low, and such a claim should be bolstered with a direct measurement of the TDD by, for example, plan-view TEM. The concern would be that etch-pits may only be revealing a fraction of the threading dislocations (TDs), and thereby providing only a lower-limit for TDD. For example, etch pits may only be revealing bunches of TDs in close spatial proximity, and not revealing isolated TDs. Etch pits may ultimately be the correct way to assess a very low TDD, but a one-to-one correspondence between TDs and etch pits needs to be solidly established for the reader. CL data is also offered, but this is also an indirect measure of TDD, and it isn't clear if CL results can discriminate between (for example) $1e6$ and $1e4 \text{ cm}^{-2}$.

The reviewer requests some data to substantiate the low threading dislocation density. In order to confirm the value claimed in this research, plan-view TEM analyses were carried-out as suggested. High quality plan-view TEM specimens were fabricated using a focused ion beam instrument. The planar sample-FIB preparation is composed of 4 steps: (i) machining a 3D piece onto the specimen surface in the selected area, (ii) the extraction of the 3D wedge containing the selected area, (iii) the planarization of the 3D form and (iv) the final cleaning step.

The threading dislocations density (TDD) can be more accurately determined from the plan-view TEM by estimating the dislocation density for a given area and repeating the process at a number of locations across the entire sample. Two samples were studied for comparison: a non-treated

Ge/Si substrate and a nanovoid-based Ge/Si substrate (NVS). From each sample, we analyze two TEM lamella taken from the edge and the centre of each substrate. The contrast in the images is dominated by thickness fringes and bend contours, which are inevitable in plan-view crystalline samples. Figure 6e and 6f below is a comparison between the TDD of two lamellae having an area of $15 \mu\text{m}^2$ each, taken from these two substrates. The estimated TDD from the non-treated Ge/Si substrate is $8.5 \pm 0.5 \times 10^8 \text{ cm}^{-2}$ compared to the estimated value of $2.5 \times 10^7 \text{ cm}^{-2}$ obtained by EPD analyses (see the table bellow). This seems to confirm the point raised by the reviewer that etch-pits may only be revealing a fraction of the threading dislocations. The difference between these two techniques has been demonstrated by Kimerling's group which shows the high correlation between the threading-dislocation densities measured by plan-view TEM and EPD [8]. For the NVS, **no threading dislocation** is found in the areas as shown in Fig. 6f. Similar observations were obtained for the other pair of lamellae.

	Non-treated Ge/Si substrate	Nanovoid-based Ge/Si substrate
plan-view TEM	$8.5 \times 10^8 \text{ cm}^{-2}$	0 cm^{-2} (within the limits of statistics)
EPD	$2.5 \times 10^7 \text{ cm}^{-2}$	$\sim 10^4 \text{ cm}^{-2}$

It is difficult, even with a plan view TEM image to measure a TDD of $\sim 10^4 \text{ cm}^{-2}$ in a quantitative manner. At that level, the average distance between dislocations is of the order of $100 \mu\text{m}$. The fact that we did not observe any dislocations on our images should confirm that the actual TDD is at least an order of magnitude lower than 10^6 cm^{-2} (average distance of $10 \mu\text{m}$). We therefore argue that the TDD of $\sim 10^4 \text{ cm}^{-2}$ obtained by the EPD technique is reasonable when compared with other results from the same technique in the literature, even though a more accurate figure remains to be defined. In order to clarify this point, we have added the following sentences in the last paragraph of section 2.2:

“The TD density can also be determined from the plan-view TEM images by estimating the dislocations in a given area at a number of zones across the entire samples [8]. Figure 6e and 6f shows two-TEM plane-view images for an untreated Ge/Si substrate and a NVS, respectively. The contrast in the images is dominated by thickness fringes and bend contours, which are inevitable in plan-view crystalline samples [9], [10]. The dark spots show dislocation strain induced bending of the thinned film. The estimated TD density from Fig. 6e is $8.5 \pm 0.5 \times 10^8 \text{ cm}^{-2}$, while no threading dislocation is found in most of the areas of the NVS as shown in Fig. 6f (within statistical limits). While the fact that TEM images on an area of $15 \mu\text{m}^2$ do not show any dislocation does not constitute a quantitative measurement, it does imply that the actual TD density must be at least an order of magnitude below a value of $\sim 10^6 \text{ cm}^{-2}$, which would correspond to an average distance of about $10 \mu\text{m}$ between dislocations. Considering the high correlation between the TD density measured by plan-view TEM and EPD [8], we argue that the $\sim 10^4 \text{ cm}^{-2}$ figure for the TD density as measured by EPD is reasonably representative and should be comparable with other results from the literature measured in the same manner. By combining different interactions between dislocations and voids, located either in the Ge layer or in the Si substrate, the averaged TD density is reduced significantly from $\sim 10^8 \text{ cm}^{-2}$ to a lower limit of $\sim 10^4 \text{ cm}^{-2}$ for $1.5 \mu\text{m}$ thick Ge layers, which is considered very low for such a thin epitaxial layer [11]–[13].”

For the reviewers' information, we include plan-view TEM of two samples: non-treated Ge/Si substrate and the NVS in Figure 6.

Figure 6. Picture of the NVS produced on a 4 in. wafer (a) and etch-pits densities taken from different regions of the NVS providing a lower-limit for TDD of $\sim 10^4 \text{ cm}^{-2}$ for 1.5 μm thick Ge layer (b-d). Plan-view TEM micrographs marking the emergence sites of dislocations on Ge films on Si taking from (e) non-treated Ge/Si sample and (f) the NVS sample. These observations confirm the TDD reducing by introducing nanovoids inside the Ge/Si substrate.

R2.2 If the TDD of $1\text{e}4 \text{ cm}^{-4}$ can be conclusively proven, then this experimental result is strong enough to merit publication with only a few comments about possible TD-removal mechanisms. Some of the discussion in the current version is insightful and

should be retained. In particular, the creation of voids could facilitate interactions between dislocations, enabling the dislocation network to change its connectivity in a way which facilitates the subsequent annihilation of TD segments. The etching of the interfacial misfit dislocation (MD) network may also play a crucial role, so this should also be mentioned as a possible enabling step unique to this process.

However, the paper would be strengthened if much of the speculation beyond this were simply removed, because the current version of the text perpetuates many misconceptions about dislocations, and does not address how the TDD at the TOP of the Ge epilayer is reduced. Perhaps the best example is Figure 1, which sets the stage for confusion throughout the remainder of the paper. The "before" image shows two dislocation half-loops, but the remainder of the TDs appear to end abruptly at the Ge/Si interface. To provide a solid basis for discussion, all TDs should be either connected pairwise with MDs to form half-loops, or continue as TDs into the Si substrate. Where MDs are concealed within the 3D image, perhaps dashed lines could be used.

In the middle panel of Figure 1, all dislocations have been omitted, so absolutely no information about what the porosification might be doing to the TD network is conveyed. In the "after" panel of Figure 1, isolated dislocations appear to terminate at voids, and this is topologically impossible. Wherever there appears to be a single dislocation entering a void, there must be at least one other dislocation entering or exiting it. To convey meaningful information, these additional dislocation segments should be included, perhaps with dashed lines if they travel through the middle of the 3D rendering. Perhaps most importantly, though, there is no illustration for how the TDs in the "before" panel have either climbed, glided or combined to reduce the TDD at the top of the Ge layer. All of the problems in the text follow from this. Therefore, my recommendation is to

a) Replace Figure 1 with a set of diagrams which correctly, completely and sequentially illustrate a process by which the TDD at the top of the Ge layer is reduced.

Thank you for all these suggestions. We have replaced the Fig. 1 taking into consideration the corrections that we summarize in:

- In the before panel, all TDs are now either connected pairwise with MDs to form half-loops, or continue as TDs into the Si substrate.
- In the middle panel, a porous interface has been created. The dashed lines indicate the TDs and MDs that have been preferentially etched.
- In the after panel, each void is now connected with two dislocations whether in the Ge layer or in the Si substrate.

Figure 1: Schematic illustration of the nanovoid based Ge/Si virtual substrate (NVS) architecture after revisions.

b) Rewrite all subsequent associated text describing this process. If this can't be done, then it would be best to provide instead a brief discussion of possible dislocation annihilation mechanisms, along with an acknowledgement that the full process is not understood at this point. This same advice applies to all of the subsequent statements about dislocations in the paper. For example (lines 274-276) "In Si, TDs bend into the glide plane towards the voids in Si onto the glide plane in order to minimize their length, causing their elimination at the nearest free surface." It should be possible to illustrate this sentence with a topologically correct sequence of diagrams, and doing this should ultimately help explain how the TDD at the top of the Ge layer is reduced. If this can't be done, consider omitting this sentence, and many other similar sentences.

As suggested, we have added a new figure (Fig. 4) in the manuscript, which schematizes possible dislocation annihilation mechanisms by the voids. For the sake of clarity, only one $\{111\}$ glide plane is drawn. Figure 4a-4d show possible TD-removal mechanisms with void located inside the Ge layer, while Figure 4e and 4f show TD-removal when the void is located in Si substrate. In order to convey this in the manuscript, the following text has been added in the discussion of "Dislocation Annihilation Mechanisms and Crystal Quality":

Figure 4. Descriptive scheme of the phenomena leading to a low threading dislocation density in the NVS. By introducing nanovoids inside the Ge layer, the dislocations with opposite Burgers vectors move easily along the $\{111\}$ glide planes during high temperature annealing (a-b) and react with each other (c), allowing the threading components to disappear (d). When the voids are located at the Ge/Si interface (on the Si side), dislocations bend towards the porous area, instead of emerging at the Ge surface causing their elimination through a similar process at the nearest free surface (e,f).

“Figure 4 proposes a descriptive sketch to explain the phenomena of the annihilation of dislocations in Ge epilayer grown on Si substrate by introducing nanovoids. For the sake of clarity, only one $\{111\}$ glide plane is drawn. The heat treatment stimulates the propagation of dislocation loops along glide planes (Fig. 4b). The dislocations with opposite Burger vectors move easily and react with each other (Fig. 4c), since it can occur in the same glide plane $\{111\}$, without the need of any extra point defect supersaturation for climbing. Thus, the threading components disappear (Fig. 4d).”

“However, the full process could be confirmed and better understood using in-situ real-time observations with transmission electron microscopy.”

“The analog is also present in Si, where TDs originating from the interface bend downwards and move along the glide plane towards the voids in Si in order to minimize their length. Their elimination in pairs is also possible within these voids. A descriptive scheme of this phenomenon

leading to the reduction of TD density when using a voided Si template is given in Figure 4e and 4f.”

“In fact, the creation of voids could facilitate interactions between dislocations, enabling the dislocation network to change its connectivity in a way which facilitates the subsequent annihilation of TD segments.”

c) Finally, a quick comment on the title. There seems to be a mismatch between prominence of "light-emitting III-V on Si" in the title and the introduction of the paper which mostly discusses III-V solar cells (and not LEDs or lasers).

Thanks for the comment. The application of this approach can indeed cover several technologies, which depend on high quality lattice mismatch materials. These include LEDs, lasers, and photovoltaic devices to name a few. In the manuscript, we have clarified this point by adding the following:

“The last decade has witnessed considerable progress in many advanced technologies based on mismatched hetero-epitaxial semiconductors such as solar cells, LED and laser sources [14]–[19].”

The title has been changed to:

“Uprooting defects to enable high-performance III-V devices on Si”

Reviewer #3

This paper reports on a novel method that allows reducing the dislocation density in lattice-mismatched hetero-epitaxy semiconductor systems. The ingenuity of the method resides on the use of a nanovoid lattice to block dislocations and thereby favor their annihilation. It is very interesting and the method was thoroughly checked by transmission electron microscopy techniques. Structure of the paper is good. The writing is relatively good, but deserves improvement here and there.

R3.1 The use of the tense of the verbs is at times awkward. For example, in the introduction it is said that ‘... have been discussed.’ for a topic that is discussed in the present paper. I suggest to use the present tense instead. The use of the verb ‘show’ should be checked (use instead e.g. ‘exhibit’, ‘present’).

Thank you for the comment. The contents of this manuscript are reviewed and corrections are included in the subsequent version.

R3.2 Figures and graphs are clear. There are however, a number of issues the authors should address before the paper can be accepted for publication. The part explaining the various dislocation mechanisms at play in their system is confused if not weak:

‘Different phenomena may come into play in the dislocation pinning mechanism, which are all promoted by high temperature achieved during annealing including; gliding, [50] climbing processes, [51,52] the void size effect[53,54] and inertial effects.

a) This is a vague statement, what is meant by ‘promoted’ in this context? does it mean ‘enhanced’? In this case one cannot state it as a generality:, i.e. temperature does not ‘promote’ all the different phenomena described here. For instance, the obstacle strength will actually decrease when temperature increases (for voids: see

Fig. 8 in Journal of Nuclear Materials 382 (2008) 147-153 and for a comparison between different types of obstacles: see Fig. 6 in Acta Materialia 64 (2014) 24-32).

Pinning of dislocations has been cited herein as a possible mechanism to describe TEM figures. We present general effects that could influence this mechanism during annealing. Indeed, the critical stress required for an edge dislocation to pass through an obstacle decreases when the temperature increases (from 10 to 700 K) according to S.M. Hafez Haghghat et al. [20], whereas it increases with increasing applied strain rate, or dislocation speed. [21]. It seems that the sentence does not reflect the intended meaning. The reviewer makes a good point that this can be written more clearly. We have rewritten it as follows:

“Different processes may come into play in the dislocation pinning mechanism including; gliding [22] and climbing processes [23], [24], which depend on temperature, strain, obstacle specifications (type, diameter and spacing) [25]–[27] and other critical parameters [21].”

b) 'void size effect': This doesn't grasp all elements needed to rationalize dislocation-void interaction: actually it is about the size and the number density of voids (this can be summarized by the distance between voids, surface to surface).

Indeed, the 'void size effect' does not cover all the parameters of the voids that influence the dislocation-void interaction such as the void diameter and the void spacing. We replaced it with:

“...obstacle specifications (type, diameter and spacing)...”

“...depends on the void parameters...”

c) 'inertial effects': This concerns the detailed dynamics of the dislocation-defect interaction. In this study these effects can be neglected, as the interest here is to consider the dislocation configuration at equilibrium. I wouldn't even mention inertial effects.

We agree with the reviewer, the inertial effect in dislocation-localized obstacle interactions could be neglected as it concerns the dynamic aspect of dislocations overcoming obstacles. In addition, the direct observation of inertial effects is only possible with atomistic simulations, which is not our interest here. As suggested, we removed this mention from the text.

d) *'For large voids that are sufficiently spaced out, dislocation can bypass the obstacle via the Orowan mechanism.'* This is wrong, this doesn't occur for voids. 'Orowan mechanism means' the formation of a dislocation loop around the obstacle. However, in this case (a void) the segment of the dislocation that meets the void will disappear in the void (making a step in the internal surface; this step has the amplitude of the Burgers vector of the dislocation). Actually, a dislocation is attracted by the internal surface of the void: as this surface is a free surface, it induces image forces on the dislocation that attract the dislocation. The dislocation therefore will not stop before the surface of the void and thus cannot form an Orowan loop around it. This absorption is well explained in the paper of Crone 2015 cited by the authors (J. C. Crone, L. B. Munday, J. Knap, Acta Mater, 2015, 101, 40-47). Besides the direct interaction of the dislocation with the void, there is also the interaction of the dislocation with the stress field induced by the void in its surroundings (see Philosophical Magazine 90 (7-8) (2010) 1075-1100). This is missed by the authors.

1/ We are aware of the excellent review by J. C. Crone et al., and agree with the reviewer's comments regarding Orowan's mechanism. Despite the phrasing of the original sentence, we certainly do not mean to suggest that the formation of a dislocation loop around the obstacle is expected. Obviously, Orowan loop is the mechanism by which a dislocation bypasses repulsive obstacles, where a dislocation largely bows out to leave a dislocation loop around the obstacle [28]. The meaning in our synthesis is the critical resolved shear stress, τ_c , required for an edge dislocation to bypass a periodic array of nanovoids that are sufficiently spaced out. According to D. J. Bacon et al., τ_c is also named τ_{orowan} , whatever the nature of the obstacle (impenetrable obstacles or voids) [29], hence the confusion of our sentence.

To avoid any confusion, we deleted the sentence that mentions Orowan loops.

2/ We agree with the reviewer that the dislocation does not form a loop around the void but it penetrates the void forming an atomic step inside the inner surface, as schematize in Figure 2. The dislocation will not disappear in the void because it is topologically impossible. The reduction of the dislocation density is explained in the rest of the text by the interaction of opposite dislocations through the voids.

Figure 2: Schematic illustration of the interaction of singular screw dislocation by one void. The exit step, characterized the initial screw dislocation, is taken up by the lower and the upper parts of the inner void surface (please see Fig. 6 of the reference [30]).

3/ The reviewer draws our attention to a significant effect that we have not cited; the stress concentrations formed around the voids. This effect was well detailed by S.M. Hafez Haghghat et al. [31]. We have considered this effect in the modifications.

On reviewing the paragraph in question, we find that it does not properly convey our meaning, and have rephrased it as follows:

“The attraction and detachment process between dislocations and voids depends on the void parameters and the distance between the partials [32]. For the smaller voids, the partials strength dominates the obstacle strength. Consequently, the dislocations cut through the small voids without being trapped. For the larger voids, the obstacle strength dominates, causing the dislocation to be pinned by the voids and to bow under the internal shear stress. In addition, the void induces a stress field in its surroundings, which strongly influences the dislocation passage depending on the geometry of the interaction [31]. The short void spacing induced-high-stress concentration increases the resolved shear stress which in turn by increasing the obstacle strength [25]. Therefore, for large voids that are widely spaced out, dislocation can bypass the obstacle,

as the stress concentration formed around the voids is low. Indeed, voids with adequate dimensions and spacing create a stable energetic well for the dislocations, strongly pinning them in place.”

R3.3 Crystallographic index notation: for a direction, it should be [] (particular) or <> (general), for a plane it should be () (particular) or {} (general). Page 10: '(x-axis along the [100])'; authors have chosen a specific <100>-type direction. From this choice, the orientations following in the text should be noted as: 'for both paired spots of 020 and 002' -> 'for both paired spots of (020) and (002)'. I suggest to change '000 spot' to 'transmitted beam spot'.

Thanks for the comment. As suggested, all notations are corrected following to:

“...for both paired spots of (020) and (002)...”

“...obtained through the (11-1) spots...”

“...from the transmitted beam spot...” instead of ‘...from the 000 spot...’

R3.4 Page 9: '*Due to the attractive force created by the population of smaller voids, the approach distance between two TDs with different Burgers reaches a low value in which the TDs begin to glide together and annihilate or fuse. These mechanisms give a reasonable explanation for the dislocations reduction by introducing nanovoids in heteroepitaxial diamond films, in which, the TDs have an interaction radius higher than the interaction radius in the conventional structure without voids, which in turn increases their recombination probability.*'

-> 'TDs have an interaction radius higher', this conclusion is dumped from high: it is not justified, it is not quantified (how many nanometers? calculated on the base of what?). The simplest conclusion one can draw in this case is that one can assume the interaction radius is the same but the dislocation density being higher the distance between them is smaller, so the probability of annihilation increases.

This is another excellent question; thank you for pointing it out. We agree with the reviewer on the first part that the interaction radius for the dislocation should be the same in the conventional structure as well as the voided structure. This should be an intrinsic property of the dislocation in the fcc system.

Taking as an example the interaction of two dislocation arms located on the same slip system (111), as can be seen in the Figure 3-a. The interaction radius is defined by a red circle. Under a driving force (e.g. a thermal stress), the dislocation reacts with other dislocations or defects that enter their field, defined by the distance R_d . In other words, threading dislocations will react with one another when they approach within a characteristic interaction distance D in conventional Ge/Si (without void) [33]; which is equal to $2R_d$.

For distant dislocations, no interaction/recombination should happen, as the separation distance between the dislocations is still higher than $2R_d$ (figure 3-b). Cyclic annealing may cause some recombination, but reports in the state-of-the-art show that threading dislocation densities below $\sim 10^6 \text{ cm}^{-2}$ has not yet been reported for continuous and thin Ge/Si layers [34].

Now, by introducing a void inside the Ge layer, both dislocations will be attracted by its free surface, which will favor their recombination (Fig. 3c).

In the voided structure, one can consider that the characteristic interaction distance required for two dislocations to interact is higher than that in the conventional structure without void $D = 2R_d$

+ 2R_v. That constitutes what we wanted to express in this paragraph. Accordingly, we have changed the sentence regarding the interaction radius to:

“These different interactions between neighboring dislocations are characterized by the physically prescribed “characteristic interaction distance...”

“...These mechanisms give a reasonable explanation for the dislocations reduction by introducing nanovoids in heteroepitaxial diamond films, in which, the characteristic interaction distance required two dislocations to interact becomes higher than that in the conventional structure without voids, which in turn increases their recombination probability (Fig. S2 Supporting Information).”

and in the discussion part by:

“The results suggest that introducing nanovoids favors recombination of TDs by increasing the characteristic interaction distance between neighboring dislocations.”

The answer to the question **R2.2** may also support the mechanism proposed here.

Figure 3: Schematic illustration of two dislocation arms located on the same slip system (111) showing that the reaction with one another is only possible if they have a minimum separation distance equal to $2R_d$ in conventional Ge/Si (a,b), and equal to $2R_d+2R_v$ in voided Ge/Si (c).

R3.5 Page 10: 'Due to its porosity, the elastic modulus of Si is reduced and the substrate can be stretched, compressed, and deformed. It is therefore expected to accommodate the mismatch of heterogeneous layers and to serve as a mechanically stretchable compliant substrate.'

Did the author checked the mechanical integrity? uthors should really check the mechanical integrity of the resulting device. Such an array of nanovoids introduces a fragilized area that may lead to the premature failure of the whole device during transport and handling. Authors should add a comment in the text about this point.

We agree with the reviewer that introducing nanoscale porosity into the virtual substrate NVS is likely to dramatically lower its stiffness and hardness. Nonetheless, we believe that robust porous silicon device and system performance will ultimately rely on achieving the appropriate

mechanical properties, in conjunction with the novel functionality achieved through nanostructuring, such as compliance and trapping of dislocations.

With regards to the comment about how the mechanical properties of the porous substrate affect the integrity of the resulting device, we have not conducted a complete study on these ourselves. Indeed, many other groups have studied the heteroepitaxy on porous silicon and devices were obtained, though most do not specifically discuss the mechanical integrity [35].

In order to test the mechanical strength of the resulting substrate and its reliability in the reduction of the dislocation density, we carried out a cyclic annealing test.

Annealing of the NVS samples was performed in a gas ($N_2: H_2$ 90:10) ambient using a J.I.P. ELEC JetFirst rapid thermal annealing system with a ramp rate of $15^\circ C s^{-1}$. The annealing temperature pulses and their duration are respectively $800^\circ C$ for 1 min and $400^\circ C$ for 1 min. The duration was fixed at 45 min, as shown in Fig. 4.

Figure 4: Schematic graph of the annealing cycle that the NVS has undergone in order to test its mechanical integrity.

After cyclic annealing, cross-section SEM image of NVS sample shows that the Ge/voided Si interface remains intact, with a slight increase of the void size in Si, due to the coalescence of small voids at high temperature (see Fig. 5).

Optical microscope inspection did not present any cracks on the top surface despite the thermal stress. This confirms that the Ge layer was not damaged or delaminated.

Figure 5: Ge/voided Si interface of the NVS sample after cyclic annealing.

The dislocation density was also verified and etch-pit density (EPD) analyses were carried-out on the NVS sample before and after the cyclic annealing using the same etching solution described in the manuscript. It has been shown by SEM images in Figure 6 that the TDD after annealing remains almost identical to that without annealing, which confirms that the dislocations remain effectively blocked from propagating in the NVS.

Figure 6: Etch pit densities taken from (a) as-prepared NVS sample and (b) after cyclic annealing.

This test does not give complete information on the mechanical integrity of our NVS substrate. However, it gives us an idea of its mechanical strength under thermal stress.

We thank the editors and reviewers for their diligent review of the manuscript. We hope they find the revisions satisfactory, and that the revised manuscript is suitable for publication in *Nature Communications*.

References:

- [1] E. A. Fitzgerald *et al.*, "Totally relaxed Ge_xSi_{1-x} layers with low threading dislocation densities grown on Si substrates," *Appl. Phys. Lett.*, vol. 59, no. 7, pp. 811–813, 1991.
- [2] K. V. and P. B. J. Scott Ward, Timothy Remo, Kelsey Horowitz, Michael Woodhouse, Bhushan Sopori, "Techno-economic analysis of three different substrate removal and reuse strategies for III-V solar cells," *Prog. photovoltaics Res. Appl.*, vol. 24, pp. 1284–1292, 2016.
- [3] T. M. Hatem and M. T. Elewa, "MODELING of dislocation evolution in Multi-junction based Photovoltaic devices," no. January, pp. 157–159, 2014.
- [4] N. P. Blanchard, A. Boucherif, Ph. Regreny, A. Danescu, H. Magoariec, J. Penuelas, V. Lysenko, J.-M. Bluet, O. Marty, G. Guillot, G. Grenet, "Engineering Pseudosubstrates with Porous Silicon Technology," in *Semiconductor-On-Insulator Materials for Nanoelectronics Applications*, 2011, pp. 47–50.
- [5] S. M. Myers and D. M. Follstaedt, "Forces between cavities and dislocations and their influence on semiconductor microstructures," *J. Appl. Phys.*, vol. 86, no. 6, pp. 3048–3063, 1999.
- [6] M. Bafleur, A. Munoz-Yague, J. L. Castano, and J. Piqueras, "Photoluminescence of molecular beam epitaxially grown Ge-doped GaAs," *J. Appl. Phys.*, vol. 54, no. 5, pp. 2630–2634, 1983.
- [7] R. Beeler *et al.*, "Comparative study of InGaAs integration on bulk Ge and virtual Ge/Si(1 0 0) substrates for low-cost photovoltaic applications," *Sol. Energy Mater. Sol. Cells*, vol. 94, pp. 2362–2370, 2010.
- [8] H.-C. Luan *et al.*, "High-quality Ge epilayers on Si with low threading-dislocation densities," *Appl. Phys. Lett.*, vol. 75, no. 19, pp. 2909–2911, 1999.
- [9] C. B. Williams, D.B., Carter, "Transmission Electron Microscopy," *Springer*, p. 369–378., 1996.
- [10] K. J. O'Shea *et al.*, "Fabrication of high quality plan-view TEM specimens using the focused ion beam," *Micron*, vol. 66, pp. 9–15, 2014.
- [11] O. pchelyakov. Yu Bolkhovityanov, "GaAs epitaxy on Si substrates: modern status of research and engineering," *Rev. Top. Probl.*, vol. 178, no. 5, p. 459, 2008.
- [12] H. Ye and J. Yu, "Germanium epitaxy on silicon," *Sci. Technol. Adv. Mater.*, vol. 15, p. 24601, 2014.
- [13] and S. A. R. C. L. Andre, A. Khan, M. Gonzalez, M. K. Hudait, E. a. Fitzgerald, J. a. Carlin, M. T. Currie, C. W. Leitz, T. a. Langdo, E. B. Clark, D. M. Wilt, "Impact of threading dislocations on both n/p and p/n single junction GaAs cells grown on Ge/SiGe/Si substrates," in *Conference Record of the Twenty-Ninth IEEE Photovoltaic Specialists Conference*, 2002, pp. 1043–1046.
- [14] F. Proulx *et al.*, "Measurement of strong photon recycling in ultra-thin GaAs n/p junctions monolithically integrated in high-photovoltage vertical epitaxial heterostructure architectures with conversion efficiencies exceeding 60%," *Phys. Status Solidi - Rapid Res. Lett.*, vol. 11, no. 2, pp. 2–6, 2017.
- [15] B. H. Le, S. Zhao, X. Liu, S. Y. Woo, G. A. Botton, and Z. Mi, "Controlled Coalescence of AlGaIn Nanowire Arrays: An Architecture for Nearly Dislocation-Free Planar Ultraviolet Photonic Device Applications," *Adv. Mater.*, pp. 8446–8454, 2016.
- [16] A. Tanaka, W. Choi, R. Chen, and S. A. Dayeh, "Si Complies with GaN to Overcome Thermal Mismatches for the Heteroepitaxy of Thick GaN on Si," *Adv. Mater.*, vol. 29, no. 38, pp. 1–6, 2017.
- [17] S. Chen *et al.*, "Electrically pumped continuous-wave III-V quantum dot lasers on silicon," *Nat. Photonics*, vol. 10, no. 5, pp. 307–311, 2016.
- [18] R. G. Beausoleil, D. Liang, M. Fiorentino, G. Kurczveil, and X. Huang, "Integrated finely tunable microring laser on silicon," *Nat. Photonics*, vol. 10, no. 11, pp. 719–722, 2016.
- [19] D. T. Spencer *et al.*, "An optical-frequency synthesizer using integrated photonics," *Nature*, vol. 557, no. 7703, pp. 81–85, 2018.
- [20] A. (2004): 161-9. Fenkçi IV, Maternal Fizyoloji. "Çiçek MN, Ed." Kadın Hastalıkları ve Doğum Bilgisi, Öncü Basımevi, R. C. Team, W. Oreza, and A. J. "ANGIOSTRONGYLUS-V. I. D. I. W. . V. R. 120. 1. (1987): 424-424. Trees, "Scholar (2)," *Profesional Psychology*, vol. 3. p. 45, 1974.

- [21] S. M. Hafez Haghghat, R. Schäublin, and D. Raabe, "Atomistic simulation of the $a_0 \langle 1\ 0\ 0 \rangle$ binary junction formation and its unzipping in body-centered cubic iron," *Acta Mater.*, vol. 64, pp. 24–32, 2014.
- [22] A. Simar, H. J. L. Voigt, and B. D. Wirth, "Molecular dynamics simulations of dislocation interaction with voids in nickel," *Comput. Mater. Sci.*, vol. 50, no. 5, pp. 1811–1817, 2011.
- [23] Y. N. Osetsky and D. J. Bacon, "Comparison of void strengthening in fcc and bcc metals: Large-scale atomic-level modelling," *Mater. Sci. Eng. A*, vol. 400–401, no. 1–2 SUPPL, pp. 374–377, 2005.
- [24] A. Dutta, M. Bhattacharya, N. Gayathri, G. C. Das, and P. Barat, "The mechanism of climb in dislocation-nanovoid interaction," *Acta Mater.*, vol. 60, no. 9, pp. 3789–3798, 2012.
- [25] J. C. Crone, L. B. Munday, and J. Knap, "Capturing the effects of free surfaces on void strengthening with dislocation dynamics," *Acta Mater.*, vol. 101, pp. 40–47, 2015.
- [26] a. Dutta, M. Bhattacharya, P. Mukherjee, N. Gayathri, G. C. Das, and P. Barat, "Anomalous interaction between dislocations and ultra-small voids," vol. 2, no. 1, 2010.
- [27] S. M. Hafez Haghghat, G. Lucas, and R. Schäublin, "Atomistic simulation of He bubble in Fe as obstacle to dislocation," *IOP Conf. Ser. Mater. Sci. Eng.*, vol. 3, 2009.
- [28] O. Wouters, *Plasticity in Aluminum Alloys at Various Length Scales*. 2006.
- [29] D. J. Bacon, U. F. Kocks, and R. O. Scattergood, "The effect of dislocation self-interaction on the orowan stress," *Philos. Mag.*, vol. 28, no. 6, pp. 1241–1263, 1973.
- [30] Y. N. Osetsky, D. J. Bacon, and V. Mohles, "Atomic modelling of strengthening mechanisms due to voids and copper precipitates in α -iron," *Philos. Mag.*, vol. 83, no. 31–34, pp. 3623–3641, 2003.
- [31] S. M. Hafez Haghghat and R. Schaublin, "Influence of the stress field due to pressurized nanometric He bubbles on the mobility of an edge dislocation in iron," *Philos. Mag.*, vol. 90, no. 7–8, pp. 1075–1100, 2010.
- [32] E. Bitzek and P. Gumbsch, "Dynamic aspects of dislocation motion: Atomistic simulations," *Mater. Sci. Eng. A*, vol. 400–401, no. 1–2 SUPPL, pp. 40–44, 2005.
- [33] S. K. Mathis, X. H. Wu, A. E. Romanov, and J. S. Speck, "Threading dislocation reduction mechanisms in low-temperature-grown GaAs," *J. Appl. Phys.*, vol. 86, no. 9, pp. 4836–4842, 1999.
- [34] Y. Yamamoto, P. Zaumseil, T. Arguirov, M. Kittler, and B. Tillack, "Low threading dislocation density Ge deposited on Si (1 0 0) using RPCVD," *Solid. State. Electron.*, vol. 60, no. 1, pp. 2–6, 2011.
- [35] L. Canham, *Handbook of Porous Silicon*. 2014.

REVIEWERS' COMMENTS:

Reviewer #2 (Remarks to the Author):

In the revisions to this paper, the authors have fully addressed one concern about the first version, and almost fully addressed the second concern. The result is a very nice manuscript about a significant result.

At this point, my only recommendation is to revise some of the statements about dislocation "pinning" and "blocking" to avoid reinforcing widely-held misconceptions. Examples of this are noted below.

Original concern 1: One concern [relates to] the threading dislocation (TDD), which, at $10e-4 \text{ cm}^2$, is remarkably low. If this is in fact the correct TDD, then this is a remarkable result which should be supported by either a direct measurement with something like plan-view TEM, or the reader should at least be provided with a solid demonstration of a one-to-one correspondence between threading dislocations (TDs) and etch pits.

Comment on current version: With the addition of plan-view TEM, this concern is now addressed.

Original concern 2: The other concern is that the explanations of possible TD-annihilation mechanisms are, at best, unintentionally misleading and incomplete. This discussion should be either (A) corrected and completed, or (B) replaced with a short discussion of a few possible mechanisms unique to this process and an acknowledgment that the full mechanism is complex and will not be fully described in this paper.

Comment on current version: This concern is largely addressed by the addition of Figure 4 and related text. Figure 4 is an excellent illustration of the most logical explanation for TD reduction. However, many potentially misleading statements from the original version still remain. With a little extra editing, these statements could all be revised to connect them to the annihilation mechanism shown in Figure 4.

This may seem overly cautious, but there are many researchers who believe that TDs can be eliminated by simply "terminating" them at a void or other internal surface. This is not the case, and persistence of this belief is a hindrance to progress. Given the effort the authors have put into this work and manuscript, and the high-profile nature of the both the result and the journal, revision of statements which appear to validate widely-held misconceptions seems worth the additional effort. Each individual instance should only require a minor revision, without affecting the flow of the surrounding text.

Examples:

Lines 39-42: Annihilation is mentioned, but the preceding sentence about "barriers", "traps" and "pinning" could be misinterpreted by many readers. This should be revised to emphasize that TD reduction is likely due to TD annihilation facilitated by nanovoids via the process shown in Figure 4. Simply "trapping" and "pinning" TDs without annihilation will not reduce the TD density.

Line 118-119: Similar comment. Readers could be led to believe that edge dislocations can be blocked by voids.

Lines 129-130: "... which acts as a free surface that blocks dislocation propagation." Readers could be misled to believe that the threading arms can simply be blocked by voids. To avoid confusion, a connection to TD annihilation should be made.

Lines 196-197, 208-215: Recommend that a sentence or two be added to the beginning of section

2.2 to emphasize the connection between dislocation "pinning" and threading arm annihilation. Perhaps explicitly stating that this is a two-step process in which multiple dislocations are attracted to the same nanovoid, facilitating the subsequent annihilation of their threading arms. Without an introductory explanation, the subsequent discussion of pinning can mislead readers who do not already understand the connection between pinning and annihilation.

Lines 300-307: This should be re-written to strengthen the connection to threading arm annihilation. A sentence about annihilation is included, but the preceding sentence will mislead many readers because it suggests that the TD density can be reduced by a buffer layer which "confines" and "traps" TDs.

Lines 315-316: The usage of the word "blocked" will mislead many readers. Either connect this thought to annihilation or remove "which confirms that...".

Line 413: Similar comment.

Line 422: Not clear what is meant by "uprooting MDs". That the etching of the material near the misfit dislocation cores is also etched? A graphic descriptor like "uproot" is appealing, but you might consider changing its usage somewhat. It would seem that it is the threading arms which are uprooted, in that the voids sever the connections between the threading arms and their associated MDs, such that half loops can form, shrink and vanish, removing pairs of threading arms.

Lines 430-432: Misleading or incorrect as written. A connection to annihilation needs to be mentioned. As with section 2.2, perhaps the beginning of section 3 should include an introductory sentence connecting pinning to annihilation.

Other technical comments:

Line 265: Is this zone amorphous, or does the XTEM beam pass through multiple laterally-shifted (by LMM accommodation) regions of crystalline material separated by voids, such that the superposition of diffraction from the different regions overlaps in a way which appears amorphous?

General comment: The long filamentary nature of the voids may play a role in the effectiveness of this technique, in that a long filamentary void could potentially capture and thereby combine many more TDs than the same void volume distributed amongst many disjoint bubbles.

Typos:

Line 245: Perhaps "The characteristic interaction distance over which two dislocations interact ..."?

Line 290: "located"

Line 392: "excited"

Line 456: "A 300 nm layer ..."

Reference 7: Seems like there was a formatting problem.

Reviewer #3 (Remarks to the Author):

The authors have duly taken into account my comments, I thank them for that. They brought the corrections that to me were necessary, and their answers to my comments or questions are fully satisfactory. I recommend the amended paper to be published without any further correction.

Responses to Reviewers:

Please find attached a revised version of our manuscript, which we would like to resubmit for publication in *Nature Communications*. The reviewers have clearly made an effort to read the paper carefully and have raised several excellent and insightful points, which have helped us to improve its quality.

In the following pages are our point-by-point responses to each of the reviewers' comments. We will refer each of the comments as, for example, 'R1.2' meaning reviewer #1's second comment.

Reviewer #2

In the revisions to this paper, the authors have fully addressed one concern about the first version, and almost fully addressed the second concern. The result is a very nice manuscript about a significant result.

At this point, my only recommendation is to revise some of the statements about dislocation "pinning" and "blocking" to avoid reinforcing widely-held misconceptions. Examples of this are noted below.

Original concern 1: One concern [relates to] the threading dislocation (TDD), which, at $10e-4 \text{ cm}^2$, is remarkably low. If this is in fact the correct TDD, then this is a remarkable result which should be supported by either a direct measurement with something like plan-view TEM, or the reader should at least be provided with a solid demonstration of a one-to-one correspondence between threading dislocations (TDs) and etch pits.

Comment on current version: With the addition of plan-view TEM, this concern is now addressed.

Original concern 2: The other concern is that the explanations of possible TD-annihilation mechanisms are, at best, unintentionally misleading and incomplete. This discussion should be either (A) corrected and completed, or (B) replaced with a short discussion of a few possible mechanisms unique to this process and an acknowledgment that the full mechanism is complex and will not be fully described in this paper.

Comment on current version: This concern is largely addressed by the addition of Figure 4 and related text. Figure 4 is an excellent illustration of the most logical explanation for TD reduction. However, many potentially misleading statements from the original version still remain. With a little extra editing, these statements could all be revised to connect them to the annihilation mechanism shown in Figure 4.

This may seem overly cautious, but there are many researchers who believe that TDs can be eliminated by simply "terminating" them at a void or other internal surface. This is not the case, and persistence of this belief is a hindrance to progress. Given the effort the authors have put into this work and manuscript, and the high-profile nature of the both the result and the journal, revision of statements which appear to validate widely-held misconceptions seems worth the additional effort. Each individual instance should only require a minor revision, without affecting the flow of the surrounding text.

We agree with the reviewer, many statements of TD reduction are now revised to connect them to the annihilation mechanism. The contents of this manuscript are reviewed and corrections are included in the subsequent version.

Examples:

Lines 39-42: Annihilation is mentioned, but the preceding sentence about "barriers", "traps" and "pinning" could be misinterpreted by many readers. This should be revised to emphasize that TD reduction is likely due to TD annihilation facilitated by nanovoids via the process shown in Figure 4. Simply "trapping" and "pinning" TDs without annihilation will not reduce the TD density.

Thank you for the comment. The words "traps" and "barriers" have been changed. In the manuscript, we have clarified this point by adding the following paragraph:

“Dislocations are eliminated from the epilayer through dislocation-selective electrochemical deep etching followed by thermal annealing, which creates nanovoids that attract dislocations, facilitating their subsequent annihilation near the Ge/Si interface.”

Line 118-119: Similar comment. Readers could be led to believe that edge dislocations can be blocked by voids.

Ok, the following sentence was removed from the text: *“In addition, strengthening effects due to the interaction between edge dislocations and voids, particularly, threading arm pinning are demonstrated.”*

Lines 129-130: "... which acts as a free surface that blocks dislocation propagation." Readers could be misled to believe that the threading arms can simply be blocked by voids. To avoid confusion, a connection to TD annihilation should be made.

Thanks, the sentence was rewritten as follows: *“... which acts as a free surface that attract dislocations, facilitating the subsequent annihilation of their threading arms.”*

Lines 196-197, 208-215: Recommend that a sentence or two be added to the beginning of section 2.2 to emphasize the connection between dislocation "pinning" and threading arm annihilation. Perhaps explicitly stating that this is a two-step process in which multiple dislocations are attracted to the same nanovoid, facilitating the subsequent annihilation of their threading arms. Without an introductory explanation, the subsequent discussion of pinning can mislead readers who do not already understand the connection between pinning and annihilation.

Thanks, as recommended, an introductory explanation has been added in the beginning of section 2.2:

“The dislocation annihilation mechanism is a two-step process in which multiple dislocations are attracted to the same nanovoid, which then leads to enhanced probability of annihilation of their threading arms.”

In addition, the statement “*The dislocation segment glides to connect void areas until its complete annihilation*” has been removed from the text.

Lines 300-307: This should be re-written to strengthen the connection to threading arm annihilation. A sentence about annihilation is included, but the preceding sentence will mislead many readers because it suggests that the TD density can be reduced by a buffer layer which "confines" and "traps" TDs.

Ok, the statement “*...and act as a buffer layer that traps and confines TDs close to the heterointerface*” has been removed from the text.

The synthesis has been rewritten as follows: “*While, the voids located at the Ge film favor the recombination of TDs and their annihilation far from the surface. TD reduction is likely due to TD annihilation facilitated by nanovoids via the process shown in Figure 4.*”

Lines 315-316 and Line 413: The usage of the word "blocked" will mislead many readers. Either connect this thought to annihilation or remove "which confirms that...".

Thanks, the word “blocked” has been replaced by “annihilated”. The statements are now written as: “*From the pit counts, the EPD is clearly lower in the voided area compared with the untreated area, which confirms that the dislocations were effectively annihilated in the case of the NVS sample.*”

“*These results confirm that the dislocations were effectively annihilated from to the epilayer surface, which is reflected by an enhancement of the emission efficiency for GaAs grown upon the NVS.*”

Line 422: Not clear what is meant by "uprooting MDs". That the etching of the material near the misfit dislocation cores is also etched? A graphic descriptor like "uproot" is appealing, but you might consider changing its usage somewhat. It would seem that it is the threading arms which are uprooted, in that the voids sever the connections between the threading arms and their associated MDs, such that half loops can form, shrink and vanish, removing pairs of threading arms.

Thanks for the interesting comment, we mean that the material around the misfit dislocation cores is also etched. As the strain energy is still stored in the Ge film even after etching, the reappearance of TD is obvious. However, the dislocation in this case passes through the formed voids, as it is the least short course (the least energy). In that case, the voids sever the connections between the threading arms and their associated MDs. It should be noted, however, that electrochemical etching also weakens the Ge/Si interface (MDs) (see Fig. 2d). The voids are also formed at the Ge/Si interface, which releases the energy accumulated at the interface by creating discontinuities in the MDs path. The reorganization of porous material and dislocations (MDs and TDs) during annealing is a simultaneous process that requires extensive studies.

The statement “uprooting MDs” is replaced by “etching MDs”.

The statement is now written as *“The results show that porous Ge is selectively formed through etching of the threading dislocation cores with higher etch rate than for the defect-free regions, up to full etching of misfit dislocations.”*

Lines 430-432: Misleading or incorrect as written. A connection to annihilation needs to be mentioned. As with section 2.2, perhaps the beginning of section 3 should include an introductory sentence connecting pinning to annihilation.

Thanks, the statement has been rewritten as following: *“In addition, the voids formed in the silicon substrate could potentially capture and thereby combine many more TDs. The TDs bend towards the voids in Si in order to minimize their length, causing their recombination and elimination at the nearest free surface thus leading to the creation of an almost defect-free Ge layer on Si.”*

Other technical comments:

Line 265: Is this zone amorphous, or does the XTEM beam pass through multiple laterally-shifted (by LMM accommodation) regions of crystalline material separated by voids, such that the superposition of diffraction from the different regions overlaps in a way which appears amorphous?

Thanks for the comment,

Several reasons may explain the appearance of a pattern diffraction of an amorphous phase in the void wall of Si:

(i) The amorphous zone appears in the voided zone, probably due to the amorphisation of the void wall during the TEM sample milling. In fact, the Ga ions used during the fabrication of TEM lamella create an amorphous nanometric layer on its surface. This amorphous layer could cover the entire void wall, as it is too thin. This postulate could be supported by analyzing the dimensions of voids in Si (>400 nm) vs. the thickness of the TEM sample (<100 nm) in Figure S2d. Taking into account these dimensions, we think that there is only one material separation between two large voids that diffracts in this zone, which is amorphous.

(ii) The superposition of diffraction from the different crystalline material separated by voids is a possibility when the dimensions of the voids are much smaller than the thickness of the TEM sample (as observed in Fig. 5h).

(iii) Incomplete recrystallization of the Si layer (the annealing temperature used for the transformation of porous Ge to nanovoids is lower than that of Si). The use of a temperature too high > 900 °C (necessary for a complete recrystallization of porous Si) causes the migration of voids in the Ge towards the surface and their disappearance (no void = low probability of TDs recombination).

To remove the ambiguity, we have reformulated the sentence according to:

“The amorphous zone appears in the void zone, probably due to the amorphisation of the void wall during the TEM sample milling or due the superposition of diffraction from the different crystalline material separated by voids.”

Typos:

Line 245: Perhaps "The characteristic interaction distance over which two dislocations interact ..."?

Line 290: "located"

Line 392: "excited"

Line 456: "A 300 nm layer ..."

Reference 7: Seems like there was a formatting problem.

Thanks for the comment. As suggested, all typos are corrected.

Reviewer #2 (Remarks to the Author):

The authors have duly taken into account my comments, I thank them for that. They brought the corrections that to me were necessary, and their answers to my comments or questions are fully satisfactory. I recommend the amended paper to be published without any further correction.

We thank again the editors and reviewers for their diligent review of the manuscript. We hope they find the revisions satisfactory, and that the revised manuscript is suitable for publication in *Nature Communications*.

Sincerely,